

# A spectacular new species of *Hyloscirtus* (Anura: Hylidae) from the Cordillera de Los Llanganates in the eastern Andes of Ecuador

Juan P. Reyes-Puig[1,2,3], Darwin Recalde[2], Fausto Recalde[2], Claudia Koch[4], Juan M. Guayasamin[5,6], Diego F. Cisneros-Heredia[3,7], Lou Jost[2,3] and Mario H. Yánez-Muñoz[2,3]

[1] Departamento de Ambiente, Fundación Oscar Efrén Reyes, Baños, Tungurahua, Ecuador
[2] Fundación Ecominga Red de Protección de Bosques Amenazados, Baños, Tungurahua, Ecuador
[3] Unidad de Investigación, Instituto Nacional de Biodiversidad (INABIO), Quito, Pichincha, Ecuador
[4] Zoologisches Forschungsmuseum Alexander Koenig, Bonn, Germany, Germany
[5] Laboratorio de Biología Evolutiva, Universidad San Francisco de Quito USFQ, Colegio de Ciencias Biológicas y Ambientales COCIBA, Instituto BIÓSFERA-USFQ, Cumbaya, Pichincha, Ecuador
[6] Department of Biology, University of North Carolina at Chapel Hill, Chapel Hill, North Carolina, United States
[7] Museo de Zoología y Laboratorio de Zoología Terrestre, Universidad San Francisco de Quito USFQ, Colegio de Ciencias Biológicas y Ambientales COCIBA, Instituto iBIOTROP, Quito, Ecuador

Corresponding author
Juan P. Reyes-Puig,
foer2005@yahoo.com

## ABSTRACT

We have discovered a spectacular new species of frog in the genus *Hyloscirtus*, belonging to the *H. larinopygion* species group. The adult female is characterized by a mostly black body with large bright red spots on the dorsal and ventral surface, extremities, and toe pads. The adult male is unknown. Small juveniles are characterized by a yellow body with variable black markings on the flanks; while one larger juvenile displayed irregular orange or yellow marks on a black background color, with light orange or yellow toe pads. Additional distinctive external morphological features such as cloacal ornamentation are described, and some osteological details are imaged and analyzed. The performed phylogeny places the new species as the sister to a clade consisting of ten taxa, all of which are part of the *H. larinopygion* group. We use genetic distances to fit the new species into a published time-calibrated phylogeny of this group; our analysis based on the published chronology suggests that the divergence of the new species from its known congeners pre-dates the Quaternary period. The new species is currently only known only from Cerro Mayordomo, in Fundación EcoMinga´s Machay Reserve, at 2,900 m in the eastern Andes of Tungurahua province, Ecuador, near the southern edge of Los Llanganates National Park, but its real distribution may be larger.

## INTRODUCTION

*Lynch & Duellman (1980)* have identified the upper Rio Pastaza watershed as a center of endemism for amphibians, and subsequent investigations have tripled the number of species apparently endemic to this region, known as the Llanganates-Sangay Ecological Corridor (*Reyes-Puig et al., 2010*, *2014*, *2015*, *2019a*, *2019b*; *Reyes-Puig & Yánez-Muñoz, 2012*; *Reyes-Puig, Reyes-Puig & Yánez-Muñoz, 2013*; *Franco-Mena, Reyes-Puig & Yánez-Muñoz, 2019*).

In the Llanganates-Sangay Ecological Corridor and the buffer zone of the Los Llanganates National Park, the Machay Reserve is a private reserve owned by the Ecuadorian NGO Fundación EcoMinga on Cerro Mayordomo. Investigators from Fundación EcoMinga and Instituto Nacional de Biodiversidad (INABIO) have been conducting botanical and herpetological expeditions there for two decades, which have led to the discovery of several dozen new species of plants, especially orchids (*Jost, 2004*) and more than ten new amphibian and reptile species (*Reyes-Puig et al., 2010*, *2014*, *2015*, *2019a*, *2019b*; *Reyes-Puig & Yánez-Muñoz, 2012*; *Reyes-Puig, Reyes-Puig & Yánez-Muñoz, 2013*; *Sheehy et al., 2014*). During a botanical expedition in March 2018, one of the participants, Darwin Recalde, fortuitously found a striking black and red frog hiding in a leaf axil of a bromeliad at eye level. During the following months and years, herpetologists from Fundación EcoMinga and INABIO conducted additional expeditions to the site and found three juveniles of the same species just a few meters from the spot where the original individual (an adult female) had been found. Further morphological and genetic comparisons identified these frogs as belonging to a new species of Stream Frog which we describe below, belonging to the genus *Hyloscirtus* Peters, 1882, in the *H. larinopygion* group.

The genus *Hyloscirtus*, in the family Hylidae, contains 38 species of arboreal frogs (*Faivovich et al., 2005*; *Frost, 2021*; *Yánez-Muñoz et al., 2021*). The genus is characterized mainly by the synapomorphy of well-developed lateral fringes on the fingers and toes (*Faivovich et al., 2005*). All known species are thought to reproduce alongside rushing streams (*Coloma et al., 2012*). The genus is distributed from Costa Rica to the Andes of Venezuela, Colombia, Ecuador, Peru and Bolivia (*Faivovich et al., 2005*; *Coloma et al., 2012*; *Frost, 2021*). The *Hyloscirtus larinopygion* group is composed of 19 species (*Frost, 2021*), of which 13 are reported from Ecuador (*Coloma et al., 2012*; *Ron, Merino-Viteri & Ortiz, 2021*). The group consists of two clades which correlate with latitude, with a small area of overlap in central Ecuador (*Almendáriz et al., 2014*; *Ron et al., 2018*). Adults of this group are characterized by a snout vent length >60 mm and dark skin color contrasting with bright patterns, especially on the arms and legs, and sometimes including the tips of the digits.

## MATERIALS AND METHODS

### Ethics statement

Our study was authorized under research permits MAE-DNB-CM-2016-0045 and MAE-DNB-CM-2019-0120, issued by the Ministerio del Ambiente del Ecuador. We followed standard guidelines for use of live amphibians and reptiles in field research

(*Beaupre et al., 2004*), compiled by the American Society of Ichthyologists and Herpetologists, the Herpetologists' League, and the Society for the Study of Amphibians and Reptiles.

## Taxon sampling

We examined specimens deposited in the herpetological collections of the Instituto Nacional de Biodiversidad, Quito (DHMECN) and Instituto de Ciencias Naturales, Universidad Nacional de Colombia, Bogotá (ICN) (Appendix 1). All museum acronyms follow *Sabaj (2016)*. Our taxonomic description employs several lines of evidence, including external morphological characters, genetic divergence, monophyly and preliminary geographic data. Similar approaches have been useful in recognizing and identifying closely related species of anurans in the eastern Andes of Ecuador (*Páez-Moscoso, Guayasamin & Yánez-Muñoz, 2011*; *Reyes-Puig et al., 2019a*, *2019b*).

The electronic version of this article in Portable Document Format (PDF) will represent a published work according to the International Commission on Zoological Nomenclature (ICZN), and hence the new names contained in the electronic version are effectively published under that Code from the electronic edition alone. This published work and the nomenclatural acts it contains have been registered in ZooBank, the online registration system for the ICZN. The ZooBank LSIDs (Life Science Identifiers) can be resolved, and the associated information viewed through any standard web browser, by appending the LSID to the prefix http://zoobank.org/. The LSID for this publication is: urn:lsid:zoobank.org:pub:4BF8C735-F06C-41AE-B130-EE41130535CC

The online version of this work is archived and available from the following digital repositories: PeerJ, PubMed Central and CLOCKSS.

## Field work

Two individuals were found fortuitously in the same spot during diurnal walks in botanical expeditions to the summit of Cerro Mayordomo (1.3702 S, 78.2679 W, 2970 m) on 16–20 March 2018 and 18–19 October 2018. Both were collected. A third individual, photographed *in situ* but not collected, was found in the same spot in December 2019, and a fourth individual was found and collected in the same area in May 2022. Several additional expeditions to the same location failed to find individuals of this species.

## Laboratory work

The two original collected individuals of the new species were taken alive, in plastic containers, to INABIO, where they were photographed in life and euthanized with benzocaine. Tissue samples were then taken for DNA sampling. They were subsequently fixed in 10% formalin for twelve hours, and then preserved as voucher specimens in 70% ethanol following the recommendations of *Heyer et al. (1994)*. These specimens were deposited in the herpetological collection (DHMECN) of INABIO as holotype and paratype. The third collected individual, an additional paratype, is being kept alive for observation and analysis at INABIO and will be deposited in the same collection.

## External morphological data

Measurements and character descriptions were made according to the specialized literature treating the *H. larinopygion* group (*Coloma et al., 2012*; *Almendáriz et al., 2014*; *Ron et al., 2018*). Description of webbing formulae of the hands and follow *Savage & Heyer (1967)* as modified by *Myers & Duellman (1982)*. We obtained morphological measurements of the two specimens preserved in 70% ethanol according to the methodology described in *Duellman (1973)*, using digital calipers (±0.01 mm). The following measurements were taken: snout-vent length (SVL), head length (HL), head width (HW), upper eyelid width (EW), interorbital distance (IOD), inter-nostril distance (IND), eye-nostril distance (END), eye diameter (ED), tympanum diameter (TD), hand length (HAL), tibia length (TL), femur length (FEL), and foot length (FL). Sex was determined by direct examination of gonads.

We also compared qualitative morphological characters between the new species and its closest relatives. Seven characters were evaluated: (1) dorsal coloration; (2) ventral coloration; (3) marks on flanks and hidden surfaces of thighs; (4) iris coloration; (5) prepollex condition; (6) in life, webbing coloration; and (7) cloacal ornamentation. Life coloration was obtained from live specimens and color photographs. Cloacal ornamentation condition was observed on live and preserved specimens.

## Osteological data and analysis

The holotype (DHMECN 14416) of the new species, and one specimen of each of five closely related species (DHMECN 12483: *Hyloscirtus lindae*; DHMECN 12111: *H. pacha*; DHMECN 6493: *H. psarolaimus*; DHMECN 3799: *H. larinopygion*; DHMECN 9686: *H. tapichalaca*), were scanned using a high-resolution micro-computed tomography (micro-CT) desktop device (Bruker SkyScan 1173; Bruker, Kontich, Belgium) at the Leibniz Institute for the Analysis of Biodiversity Change—Museum Koenig (LIB Bonn, Germany). To avoid movements during scanning, the specimens were placed in a small plastic container and mounted with styrofoam. The scans were conducted over 180 degrees with rotational steps of 0.3–0.4 degrees, with a source voltage of 35 kV and source current of 150 μA, without the use of a filter, at an image resolution of 39.3–50.0 μm. Scan duration was 30:01–45:37 min with an exposure time of 280 ms and average rate of 5 frames per second. The micro-CT datasets were reconstructed using N-Recon software (Bruker MicroCT, Kontich, Belgium) and rendered in three dimensions through the aid of CTVox for Windows 64 bits version 2.6 (Bruker MicroCT, Kontich, Belgium). Additionally, the skull of the holotype of the new species was rendered and segmented to separate and color individual bones in three dimensions using Amira visualization software (FEI, Thermo Fisher Scientific, Waltham, MA, USA). Osteological terminology follows *Trueb (1973)*, *Duellman & Trueb (1994)*, *Coloma et al. (2012)*, *Kunisch et al. (2021)*, *Reyes-Puig et al. (2021)*. For the description of the cranium and the osteology of the hand, we followed the proposal of *Coloma et al. (2012)*. Cartilage structures were omitted from the osteological descriptions, because micro-CT does not render cartilage. To facilitate comparisons among skull bones, we added color to the micro-CT scan images using Adobe Photoshop.
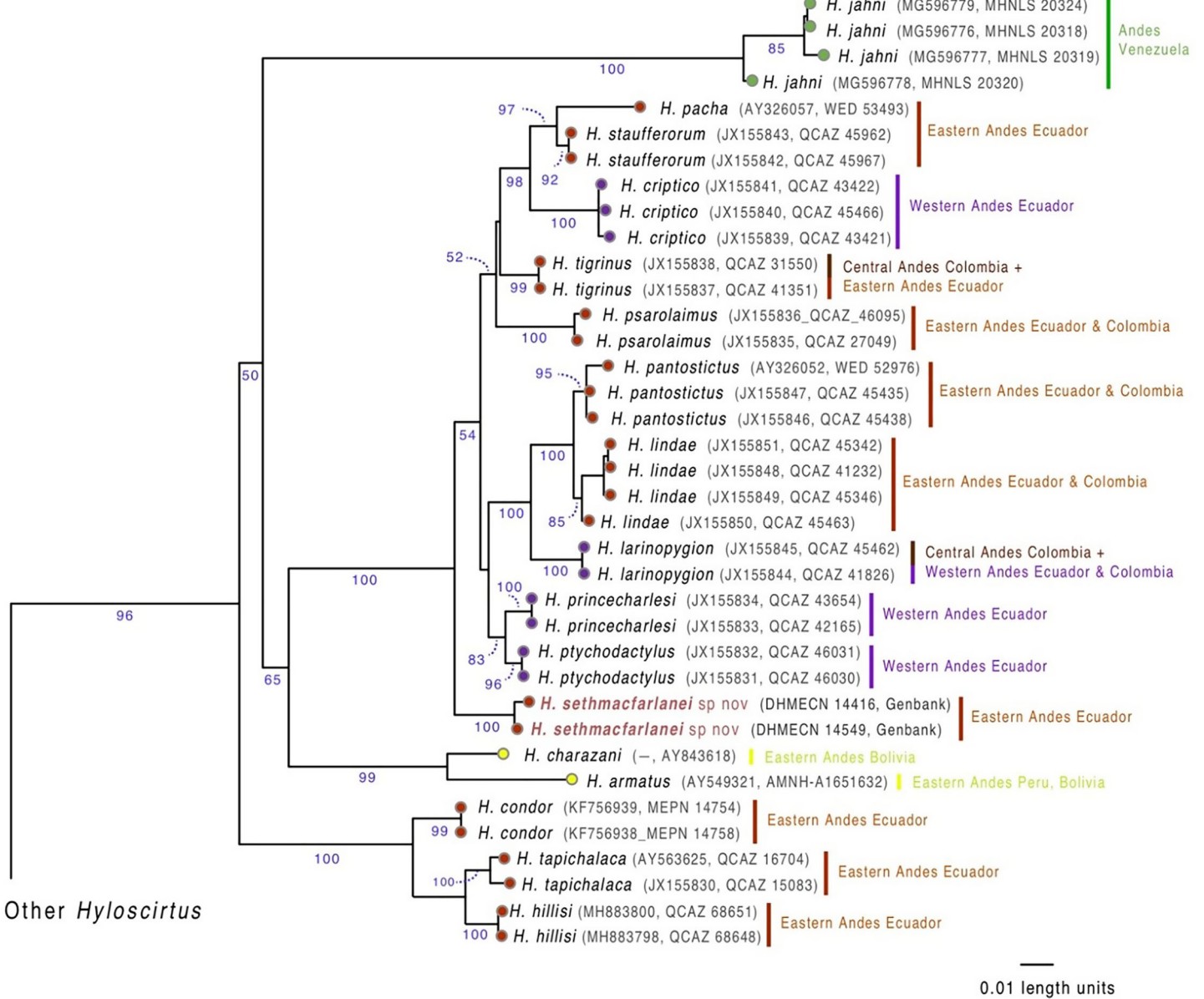

**Figure 1 Evolutionary relationships of species in the *Hyloscirtus larinopygion* group, based on the mitochondrial gene 16S under ML criterion.** Clade support (bootstrap %) are in blue. The new species is in red.

## Genetic sampling

We generated two new sequences, one from the individual collected in March 2018 (holotype DHMECN 14416) and the other from the individual collected in October 2018 (paratype DHMECN 14549), for the mitochondrial 16S gene (see Fig. 1), following the primers and protocols described in *Guayasamin et al. (2015)*. The new sequences (GenBank OM293945, OM293945) were aligned with all sequences available for *Hyloscirtus* in GenBank (http://www.ncbi.nlm.nih.gov/genbank), originally published by *Faivovich et al. (2005)*, *Coloma et al. (2012)*, *Almendáriz et al. (2014)*, *Guayasamin et al. (2015)* and *Ron et al. (2018)*. GenBank codes of downloaded species are shown in Fig. 1.

## Phylogenetic analysis

Sequences were aligned using MAFFT v. 7 (*Katoh & Standley, 2013*) with the Q-INS-i strategy. Maximum likelihood (ML) trees were estimated using GARLI 2.01 (Genetic Algorithm for Rapid Likelihood Inference; *Zwickl, 2006*). GARLI uses a genetic algorithm that finds the tree topology, branch lengths and model parameters that maximize ln(L) simultaneously (*Zwickl, 2006*). In order to determine the outgroups for our analyzes, we conducted preliminary runs in GARLI and selected as outgroups those species that were inferred as most phylogenetically distant to the *Hyloscirtus larinopygion* group. With this information, we selected species in the *H. bogotensis* group as outgroups. During the ML analyses, individual solutions were selected after 10,000 generations with no significant improvement in likelihood, with the significant topological improvement level set at 0.01. The final solution was selected when the total improvement in likelihood score was lower than 0.05, compared to the last solution obtained. Default values were used for other GARLI settings, as per recommendations of the developer (*Zwickl, 2006*). Bootstrap support was assessed *via* 1,000 pseudoreplicates under the same settings used in tree search. Genetic distances (uncorrected p) between the new species and its closest relatives were calculated using PAUP v.4.0a (*Swofford, 2002*).

## Divergence time estimation

The hybrid time-calibrated phylogeny of *Coloma et al. (2012)* for the *H. larinopygion* group, combining external fossil dates and estimated substitution rates, was used to calculate a regression line and correlation coefficient between genetic distances and divergence times for species-pairs in this group. The smallest estimated genetic distance between our new species and the other species in this group can then be inserted into this linear equation to estimate the minimum divergence time between the new species and the other sequenced species in the group. Approximate uncertainties in our estimate of divergence time were based on the uncertainty bars in Fig. 5 of *Coloma et al. (2012)*.

## Ecological niche modeling

We use Maxent (version 3.4.2) to obtain a model of the range of ecological niches for the northern clade of the *H. larinopygion* group. Localities for all species of the group were obtained from literature and museum collections. Recommended default values were used for convergence threshold, maximum number of iterations, and maximum background points; 25% of localities were randomly set aside as test points; regularization was set to 1. Selected format for representation of probabilities for models was logistic. Parametrization was based on WorldClim (version 2.1, *Fick & Hijmans, 2017*). Statistical analyses of variable contributions for data layers, including jackknife tests and correlation tests, were used to obtain more informative and less correlated variables. Models were evaluated through quantitative and qualitative tests, including threshold-independent test, threshold-dependent test, visual evaluations, and evaluation of variable importance and response curves. A geographical information system was developed based on grids from Maxent with ArcGis Desktop to analyze data and produce relevant maps.

## RESULTS

### Phylogenetic relationships

Our phylogenetic analysis (Fig. 1) shows that the new species is sister to a clade containing ten *Hyloscirtus* species: *H. criptico* (*Coloma et al., 2012*), *H. larinopygion* (*Duellman, 1973*), *H. lindae* (*Duellman & Altig, 1978*), *H. pacha* (*Duellman & Hillis, 1990*), *H. pantostictus* (*Duellman & Berger, 1982*), *H. princecharlesi Coloma et al. (2012)*, *H. psarolaimus* (*Duellman & Hillis, 1990*), *H. ptychodactylus* (*Duellman & Hillis, 1990*), *H. staufferorum* (*Duellman & Coloma, 1993*), and *H. tigrinus* (*Mueses-Cisneros & Anganoy-Criollo, 2008*). However, we note that support for the exact topology of this relationship is low (bootstrap = 54%).

Genetic distances (mitochondrial 16S percent differences calculated from uncorrected *p* values) between the new species and the most closely related *Hyloscirtus* are given in Table 1. The lowest values of genetic distances between the new species and its relatives were 2.2–2.9% to *H. tigrinus* and 2.6–2.8% to *H. ptychodactylus* (Table 1).

Most DNA sequences are publicly available (see GenBank codes in Fig. 1). The sequences of the new species are available as Supplemental Files.

### Estimation of divergence time

We confirmed our theoretical expectation that the fossil-calibrated divergence times of sequenced species-pairs within the *H. larinopygion* group (*Coloma et al., 2012*) were linearly correlated with their genetic distances (Fig. 2), with $R^2$ = 0.92 (regression line forced to go through origin) or 0.94 (unconstrained regression line). The latter correlation equation is y = 2.74x + 2.17 where x is the genetic distance (expressed as a percent) and y is the divergence time (expressed in millions of years). *Hyloscirtus sethmacfarlanei* sp.nov. was not known at the time of that study, but we can use this regression line with our measured genetic distances to estimate the divergence times between *H. sethmacfarlanei* sp. nov. and its relatives. Based on the mean genetic distance of 2.55 % between the new species and its least-distant relative *H. tigrinus*, we estimate that the divergence time between the new species and the other sequenced species is approximately 9.1 Mya +/− 4 My.

### Systematic account

*Hyloscirtus sethmacfarlanei* **sp. nov.**

**Proposed standard Spanish name:** Rana de torrente de Seth MacFarlane
**Proposed standard English name:** Seth MacFarlane's torrent frog
urn:lsid:zoobank.org:pub:4BF8C735-F06C-41AE-B130-EE41130535CC

**Holotype (Figs. 3, 4).** DHMECN 14416, adult gravid female, collected in the Machay Reserve of Fundacion EcoMinga, Cerro Mayordomo (1.370204 S, 78.267943 W, 2,970 m), Rio Verde parish, Baños township, Tungurahua province, Republic of Ecuador, on 17 March 2018, by Darwin Recalde, Fausto Recalde, Santiago Recalde, and Jordy Salazar.

**Table 1 Genetic distances (mitochondrial 16S) between *Hyloscirtus sethmacfarlanei* sp. nov. and its most closely related congeners.**

| | H. condor (n = 2) | H. hillisi (n = 5) | H. pacha (n = 1) | H. larinopygion (n = 2) | H.lindae (n = 4) | H. pantostictus (n = 3) | H. tapichalaca (n = 2) | H. ptychodactylus (n = 2) | H. princecharlesi (n = 2) | H. psarolaimus (n = 2) | H. tigrinus (n = 2) | H. staufferorum (n = 2) | H. sethmacfarlanei sp nov (n = 2) |
|---|---|---|---|---|---|---|---|---|---|---|---|---|---|
| H. condor | 0.0 | | | | | | | | | | | | |
| H. hillisi | 3.8–4.1 | 0.0–0.1 | | | | | | | | | | | |
| H. pacha | 10.2–10.5 | 11.3–11.4 | 0.0 | | | | | | | | | | |
| H. larinopygion | 9.6–9.8 | 10.2–10.4 | 4.6 | 0.0 | | | | | | | | | |
| H. lindae | 10.1–10.8 | 11.0–11.7 | 4.6–5.3 | 2.6–3.1 | 0.0–0.5 | | | | | | | | |
| H. pantostictus | 10.4–10.8 | 11.1–11.5 | 4.7–4.8 | 2.7–2.8 | 6.0–14.0 | 0.0–0.1 | | | | | | | |
| H. tapichalaca | 3.7–4.0 | 2.7–3.1 | 11.0–11.1 | 10.1–10.4 | 10.9–11.4 | 11.1–11.4 | 0.6 | | | | | | |
| H. ptychodactylus | 9.5–9.6 | 10.2–10.5 | 3.8 | 2.9–3.0 | 2.9–3.6 | 3.1–3.2 | 10.1–10.2 | 0.0 | | | | | |
| H. princecharlesi | 9.7–9.9 | 10.8–11.0 | 4.3 | 3.2 | 3.3–3.6 | 3.4–3.6 | 10.6–10.7 | 1.3 | 0.0 | | | | |
| H. psarolaimus | 10.8–11.1 | 11.3–11.7 | 4.2–4.5 | 4.8–5.0 | 4.5–5.4 | 4.8–5.2 | 11.2–11.4 | 3.6–3.8 | 3.8–4.1 | 0.3 | | | |
| H. tigrinus | 9.8–10.1 | 10.5–10.9 | 3.1–3.2 | 3.9–4.0 | 3.9–4.6 | 4.0–4.2 | 10.5–10.8 | 2.6 | 2.8–2.9 | 3.0–3.3 | 0.0 | | |
| H. staufferorum | 9.6–10.1 | 10.6–11.1 | 1.8 | 4.2–4.4 | 4.0–4.7 | 4.1–4.3 | 10.6–10.8 | 2.8–2.9 | 3.3–3.4 | 3.4–3.7 | 2.7–2.8 | 0.0 | |
| H. sethmacfarlanei sp. nov. | 9.3–9.5 | 9.2–9.6 | 3.7–3.9 | 2.9–3.5 | 3.7–4.6 | 3.5–3.7 | 9.4–9.9 | 2.6–2.8 | 3.1–3.3 | 3.1–3.3 | 2.2–2.9 | 3.5–4.0 | 0.4 |

**Note:**
Values are presented as percent distances calculated from uncorrected *p* values.

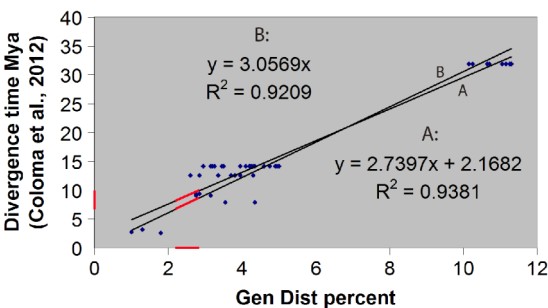

**Figure 2 Linear regression of divergence time *vs* genetic distance for some *Hyloscirtus* species of the *Hyloscirtus larinopygion* group.** Line (A) is the unconstrained regression line. Line (B) is constrained to pass through the origin. Red segments represent the ranges of genetic distances (2.2–2.9%) and inferred divergence times between *H. sethmacfarlenei* sp. nov.and *H. tigrinus*, the known species with the smallest genetic distance to the new species. The estimated divergence time should be considered only a rough estimate.

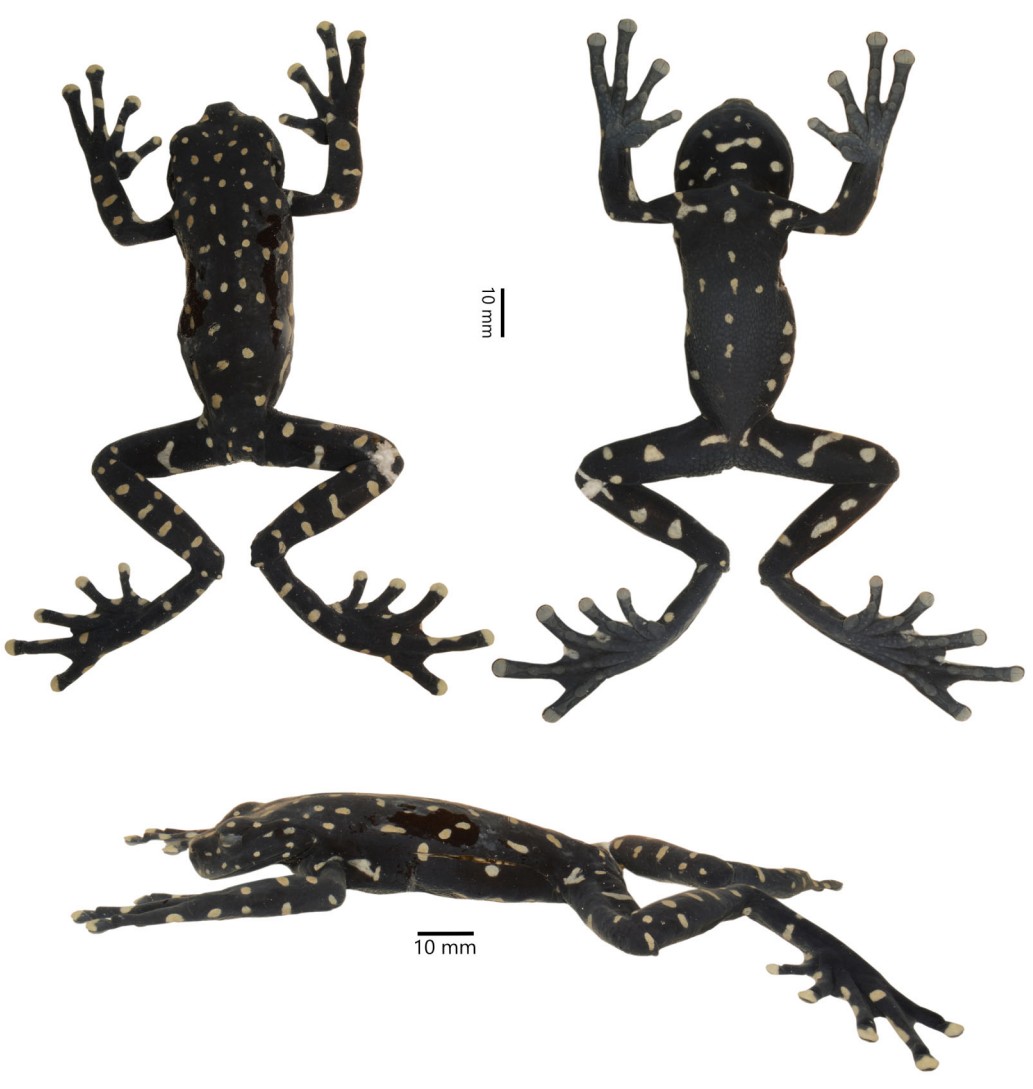

**Figure 3 Dorsal, lateral, and ventral views of the preserved holotype of *Hyloscirtus sethmacfarlanei* sp. nov. (DHMECN 14416).** Photographs: MYM.

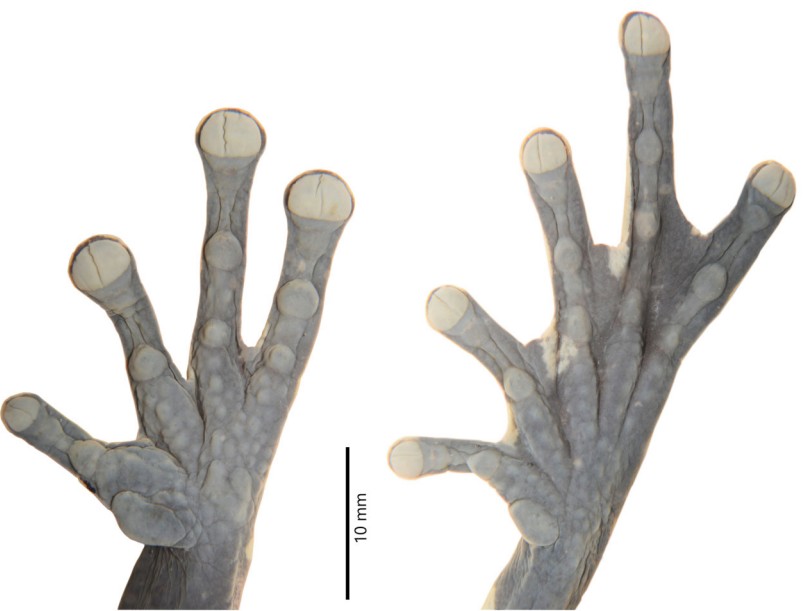

**Figure 4 Details of the hand and foot of the preserved holotype of *Hyloscirtus sethmacfarlanei* sp. nov. (DHMECN 14416).** Photographs: MYM.

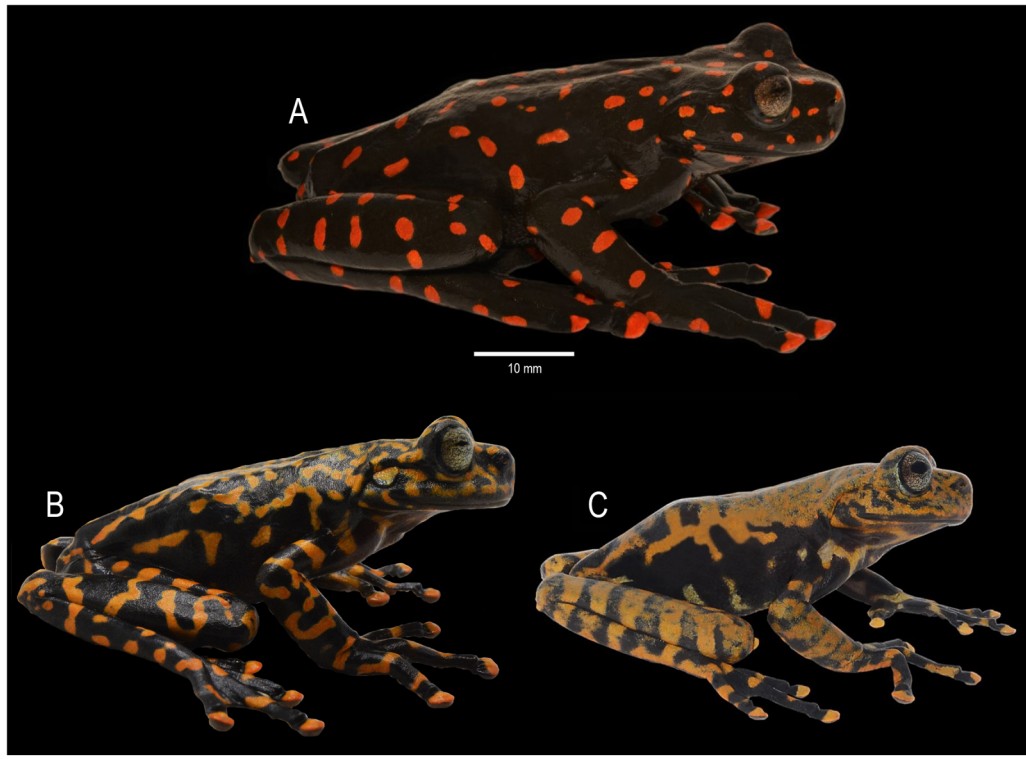

**Figure 5 Dorsal coloration of *Hyloscirtus sethmacfarlanei* sp. nov.** (A) Female holotype (DHMECN 14416). (B) Juvenile paratype (DHMECN 17554). (C) Juvenile paratype (DHMECN 14549). Photographs: MYM.

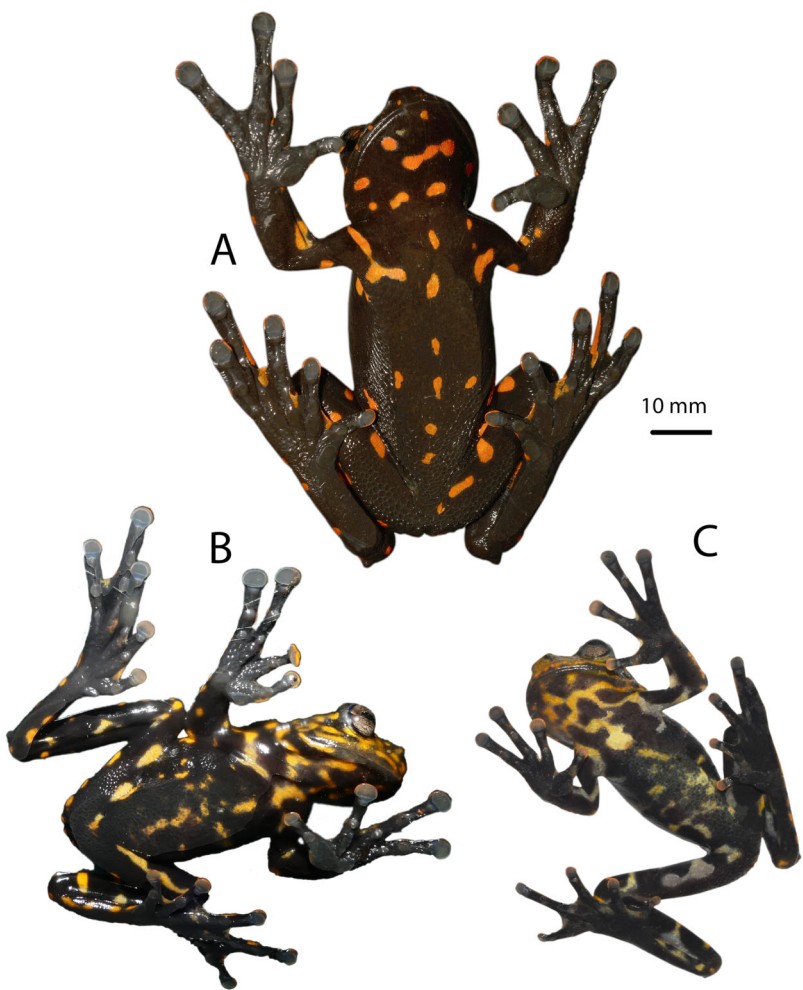

**Figure 6 Ventral coloration of _Hyloscirtus sethmacfarlanei_ sp. nov.** (A) Female holotype (DHMECN 14416). (B) Juvenile paratype (DHMECN 17554). (C) Juvenile paratype (DHMECN 14549). Photographs: MYM.         

**Paratypes (Figs. 5–8).** DHMECN 14549, juvenile, sex undetermined, without differentiation of external and internal sexual characters, collected at the type locality on 19 October 2018, by Fausto Recalde, Santiago Recalde, Darwin Recalde and Jordy Salazar; DHMECN 17554, juvenile sex undetermined, no differentiation of external and internal sexual characters, collected at the type locality on 30 May 2022, by Fausto Recalde, Luis Recalde, and Santiago Recalde.

**Generic placement.** We assign the new species to the genus _Hyloscirtus_ Peters, 1882, defined according to _Faivovich et al. (2005)_ and _Rojas-Runjaic et al. (2018)_, and to the _H. larinopygion_ group (_sensu Duellman & Hillis, 1990_; _Faivovich et al., 2005_), according to its phylogenetic position (Fig. 1) and morphological traits such as wide dermal fringes on fingers and toes, hands and legs with large terminal discs and a reduced membrane, adults characterized by a snout vent length >60 mm, and dark overall skin color contrasting with bright color patterns.

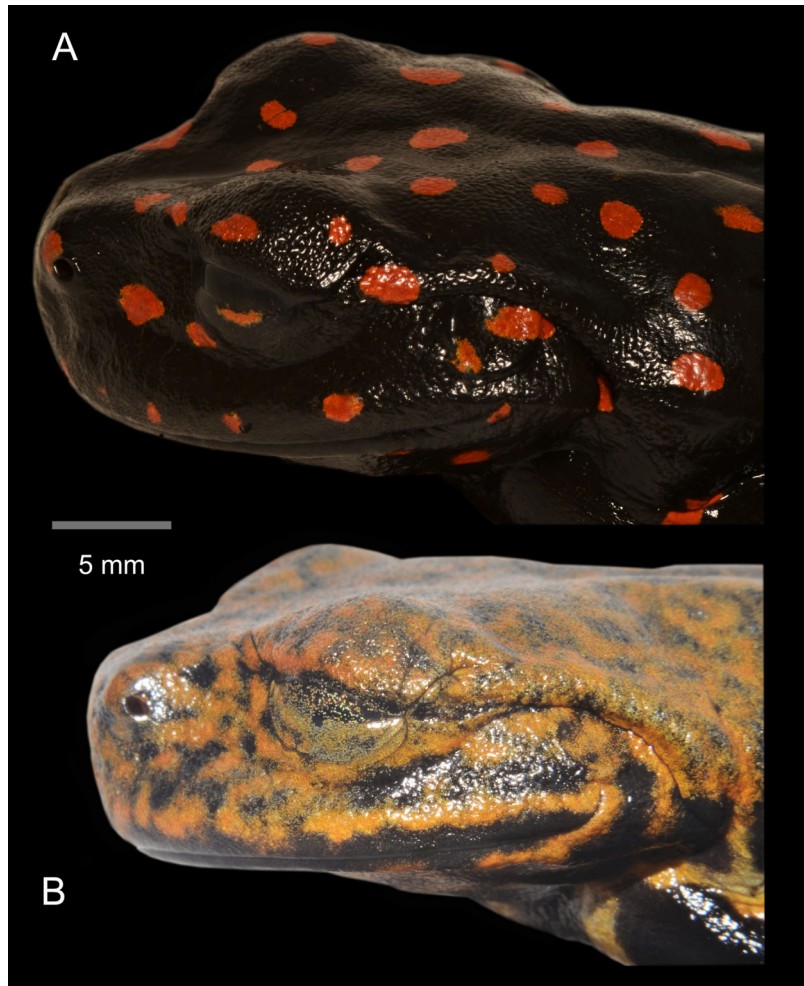

**Figure 7 Lateral detail of head in life of the type series of *Hyloscirtus sethmacfarlanei* sp. nov.**
(A) Female holotype (DHMECN 14416). (B) Juvenile paratype (DHMECN 14549). Juvenile paratype
(DHMECN 14549), note coloration in nictitating membrane. Photographs: MYM.

**Diagnosis**. *Hyloscirtus sethmacfarlanei* sp. nov. is a member of the *Hyloscirtus larinopygion* group as diagnosed by *Duellman & Hillis (1990)*, *Faivovich et al. (2005)* and *Weiens et al. (2005)*, and differs from other members of the group by the following combination of characters: discs of digits narrow; fleshy calcar present cloacal ornamentation with two thick well-defined parallel paracloacal grooves; a well-defined supracloacal fold reaching the vent; skin surrounding cloaca strongly areolate and granular; anterior border of sphenethmoid not in contact with nasal; nasal not in contact with maxilla; frontoparietals rugose; vomers not in medial contact, and with 12–13 tooth loci; 54–56 maxillary tooth loci; 10–11 premaxillary tooth loci; zygomatic ramus of squamosal slightly longer than otic ramus, and otic ramus not in contact with prootic. The adult female, in its reproductive stage with internal sexual characters defined, is further characterized by black ground color covered with large bright red spots on both the dorsal and ventral surfaces, and red tips on all digits.

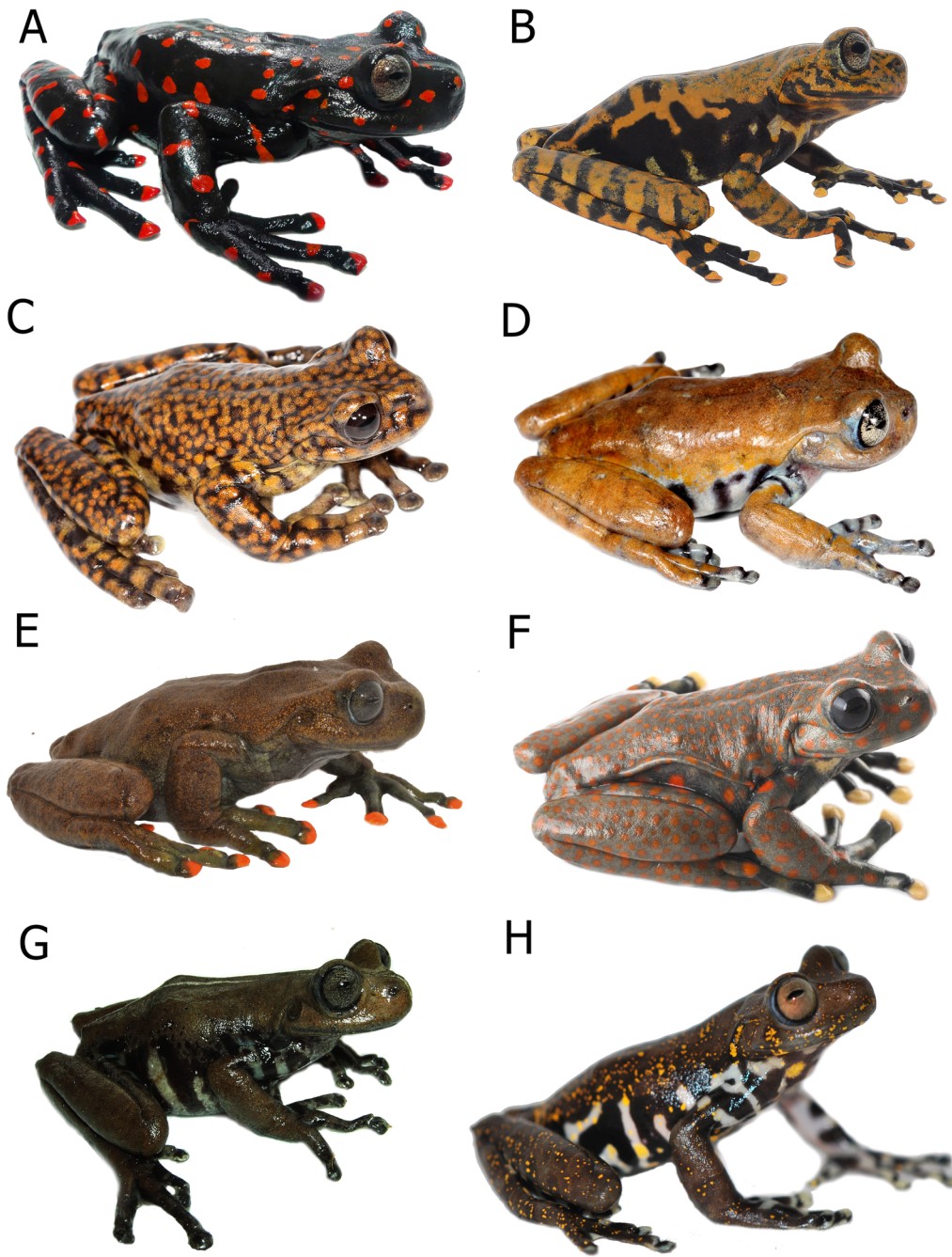

**Figure 8 Comparison in life of *Hyloscirtus sethmacfarlanei* sp. nov. with six species of *Hyloscirtus* from the *H. larinopygion* group from the Andes of Ecuador.** (A) *Hyloscirtus sethmacfarlanei* sp. nov., female holotype (DHMECN 14416). (B) *Hyloscirtus sethmacfarlanei* sp. nov., juvenile paratype (DHMECN 14549). (C) *Hyloscirtus princecharlesi* (photographic record QCAZ). (D) *Hyloscirtus larinopygion*, photographic record QCAZ. (E) *Hyloscirtus lindae* (DHMECN 12483 ). (F) *Hyloscirtus pantostictus* (photographic record QCAZ). (G) *Hyloscirtus psarolaimus* (DHMECN 6493). (H) *Hyloscirtus pacha* (DHMECN 12111). Photographs: JPRP, MYM, Santiago Ron. QCAZ, Jorge Brito.               

**Description of holotype (Figs. 3, 4).** Adult female, SVL 72.0 mm. Body slender, head rounded in dorsal view, longer than wide (head length 113% of head width); width of upper eyelid 72% of the interorbital distance; texture of the dorsal surface of the head rough, including the eyelids; snout truncate in dorsal and lateral views; eye-nostril distance slightly less than the diameter of the eye; canthus rostralis short and slightly rounded, loreal region slightly concave; internarial region flat and slightly depressed; top of head slightly concave; nostrils oval and slightly protuberant, directed laterally; eyes large and protuberant, 25% of head length; interorbital region concave; eye diameter 1.8 times larger than the diameter of the tympanic ring; supratympanic fold well-defined, directed obliquely from the posterior border of the eye, covering the dorsal edge of the tympanum, extending back to the upper shoulder; tympanum and tympanic ring evident and round, 57% of eye diameter, separated from the eye by a distance 1.6 times larger than the diameter of the tympanum.

Anterior and posterior extremities slim. Relative length of fingers I < II < IV < III; fingers with large oval disks slightly wider than finger; subarticular tubercles simple and enlarged, round and prominent; multiple round and oval supernumerary tubercles present; thenar tubercle large and flat, oval and elongated; palmar tubercle asymmetric with a slightly heart-shaped outline; prepollex absent; glandular nuptal pad covering the outer margin of Finger I; fingers long with interdigital webbing basally and with fleshy lateral fringes on all fingers.

Hind limbs long and slender, tibia length 46% of SVL; foot length 46% of SVL; heel tubercles large and round in outline; inner tarsal fold absent; large rounded to slightly oval subarticular tubercles in all fingers, supernumerary foot tubercles rounded, low; toes long, narrower than the disc, discs not expanded; relative lengths of toes I < II < V< III < IV; inner metatarsal tubercle large, oval; outer metatarsal tubercle absent; toes with interdigital membrane, toe membrane formula: I 2-3 II 3- 2 III 3-2 IV 3-2 V (Fig. 4).

Body skin is finely granular, especially on flanks; inguinal glands absent; ventral skin densely areolate, less so towards the throat. Supracloacal flap transversal, well-defined, with supracloacal fold present, reaching the level of the vent, with two paracloacal folds; skin around the cloaca strongly areolate and granular (Figs. 9, 10). Choana large and oval, notably separated from each other and perpendicular to the floor of the mouth; dentigerous processes of vomers transverse, with vomerine teeth numbering 9–10; tongue wide and rugose, slightly rounded, partially attached to the floor of the mouth.

**Coloration of holotype in life (Figs. 5–8, 11).** Entire dorsal and ventral surfaces of the head, body, and limbs black with large bright red round to oval spots scattered over the whole body, including the tips of the digits; spots 3–4 mm in diameter on dorsal surface of body and 5–10 mm long on ventral surface and throat, more elongated on the extremities and flanks. Iris light grayish with fine dark reticulations, while the nictitating membrane, revealed in defense and at rest, is well-developed, black in color, with irregular red reticulations.

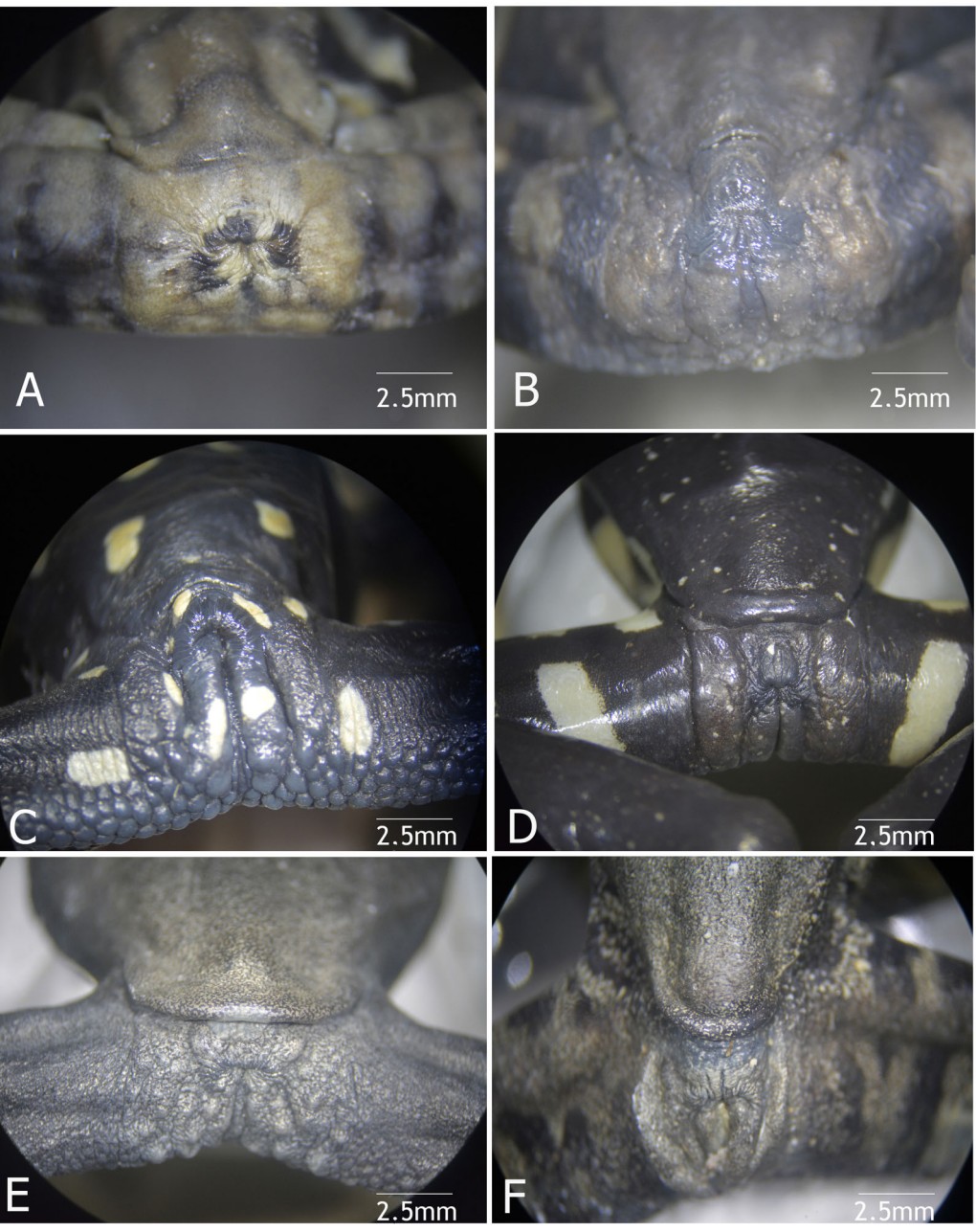

**Figure 9 Vent condition in preserved specimens of six species of *Hyloscirtus* in the *H. larynopygion* group.** (A) *Hyloscirtus larinopygyion* (DHMECN 3799). (B) *Hyloscirtus psarolaimus* (DHMECN 6493). (C) *Hyloscirtus sethmacfalanei* sp. nov. (DH MECN 14416). (D) *Hyloscirtus pacha* (DHMECN 12111). (E) *Hyloscirtus lindae* (DHMECN 12483). (F) *Hyloscirtus tapichalaca* (DHMECN 9686). Photographs: MYM.

**Coloration of holotype in preservative (~70% ethanol) (Fig. 3).** Mainly black background; the red spots in life fade to yellowish-white or white; ventral surfaces and throat grayish black with scattered irregular white elongated spots; palms of hands and feet grayish.

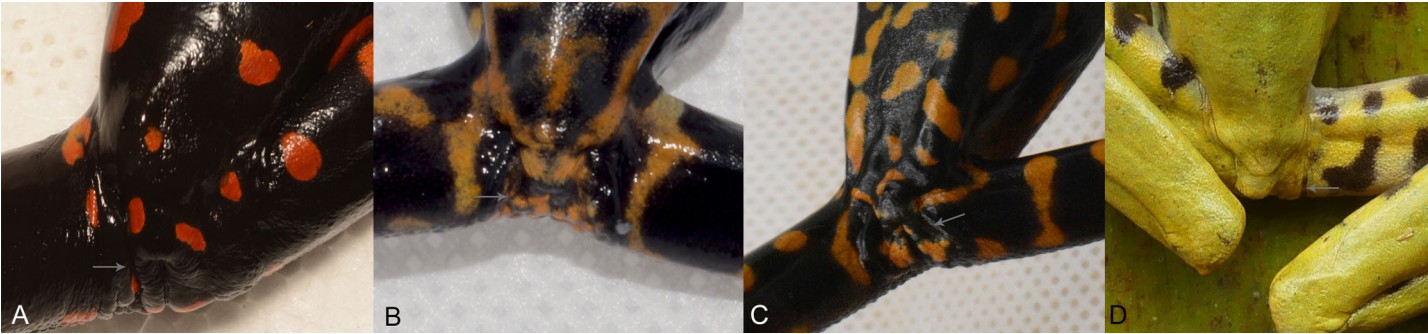

**Figure 10 Cloacal ornamentation detail in life of the type series of *Hyloscirtus sethmacfalanei* sp. nov.** (A) Female holotype (DHMECN 14416). (B) Juvenile paratype (DHMECN 14549 ). (C ) Juvenile paratype ( DHMECN 17554 ). (D) Uncollected juvenile. Gray arrows shows paracloacal folds in relation with the supracloacal fold and the vent. No scale to facilitate comparisons between different sizes of the specimens.

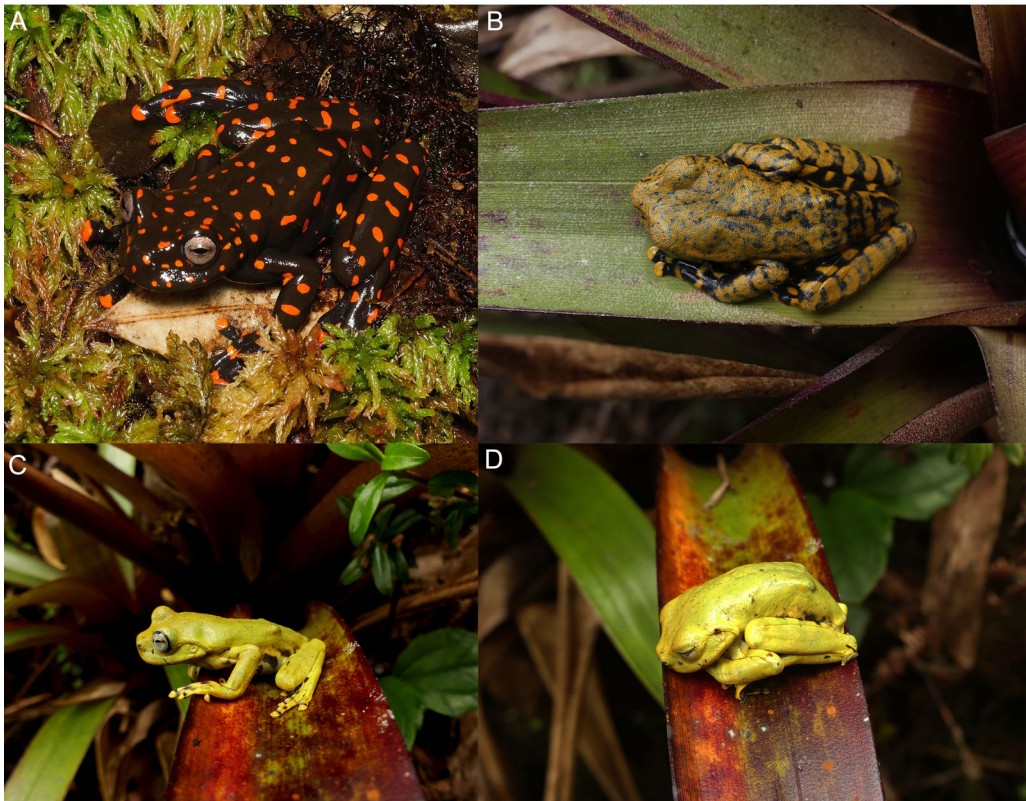

**Figure 11 Live individiuals of *Hyloscirtus sethmacfarlanei* sp. nov. *in situ*.** (A) Female holotype (DHMECN 14416). (B) Juvenile paratype (DHMECN 14549). (C) Juvenile, not collected, in its natural habitat. (D) Same juvenile adopting defensive behavior. Photos: LJ, FR, JPRP.

**Measurements of the holotype (in mm).** SVL = 72.0; HL= 22.9; HW = 20.2; EW = 6.0; IOD = 8.3; IND = 5.2; END = 5.4 ; ED = 5.7; TD = 3.2; HAL = 25.2; TL = 33.3; FEL = 26.1; FL = 33.4.

**Table 2 Morphological measurement of the type series of *Hyloscirtus sethmacfarlanei* sp. nov.**

| Morphological Measurements | Holotype adult female DHMECN 14416 | Paratype juvenile DHMECN 14549 | Paratype juvenile DHMECN 17554 |
|---|---|---|---|
| SVL | 72.0 | 46.5 | 53.9 |
| HL | 22.9 | 16.1 | 19.5 |
| HW | 20.2 | 15.6 | 18.4 |
| EW | 6.0 | 4.6 | 6.7 |
| IOD | 8.3 | 5.2 | 4.6 |
| IND | 5.2 | 3.7 | 4.8 |
| END | 5.4 | 3.5 | 3.3 |
| ED | 5.7 | 4.1 | 5.4 |
| TD | 3.2 | 2.0 | 3.1 |
| HAL | 25.2 | 16.7 | 21.6 |
| TL | 33.3 | 24.3 | 29.1 |
| FEL | 26.1 | 20.8 | 26.4 |
| FL | 33.4 | 22.7 | 28.1 |

Note:
SVL, snout-vent length; HL, head length; HW, head width; EW, upper eyelid width; IOD, interorbital distance; IND, inter-nostril distance; END, eye-nostril distance; ED, eye diameter; TD, tympanum diameter; HAL, hand length; TL, tibia length; FEL, femur length; FL, foot length.

**Measurements of the paratypes.** See Table 2.

**Osteology of the preserved holotype (Figs. 12–15).** *Coloma et al. (2012)* provide a detailed description of the osteology of the *H. larinopygion* group. In order to avoid redundancy, in the following we describe only those osteological features of the holotype of *H. sethmacfarlanei* sp. nov. where we found differences from the other species.

*Skull* (Figs. 11, 12). The anterior border of the sphenethmoid is not in contact with the nasal; the nasal is not in contact with the maxilla; the frontoparietals are rugose; the paired vomers bear 12-13 tooth loci and are not in contact medially; there are 54-56 tooth loci on each maxilla and 10-11 tooth loci on each premaxilla; the zygomatic ramus of the squamosal is slightly longer than the otic ramus, and the latter is not in contact with the prootic.

    *Posteromedial processes of the hyobranchium* (Fig. 15). The posteromedial processes of the hyobranchium are paired ossified structures, longer than broad, the anterior portion with triangular "head-like" shape, and a posterior elongated stem.

**Tadpole.** Not known.

**Advertising call.** Not known.

**Variation (Figs. 5, 6, 10 and 11).** Standard measurements from the three collected individuals are shown in Table 2. The three known juveniles (DHMECN 14549, DHMECN 17554, and the uncollected individual) share the distinctive cloacal ornamentation and skin texture of the holotype, but differ from the female holotype as follows: sexual characters not clearly evident (based on the observation of internal

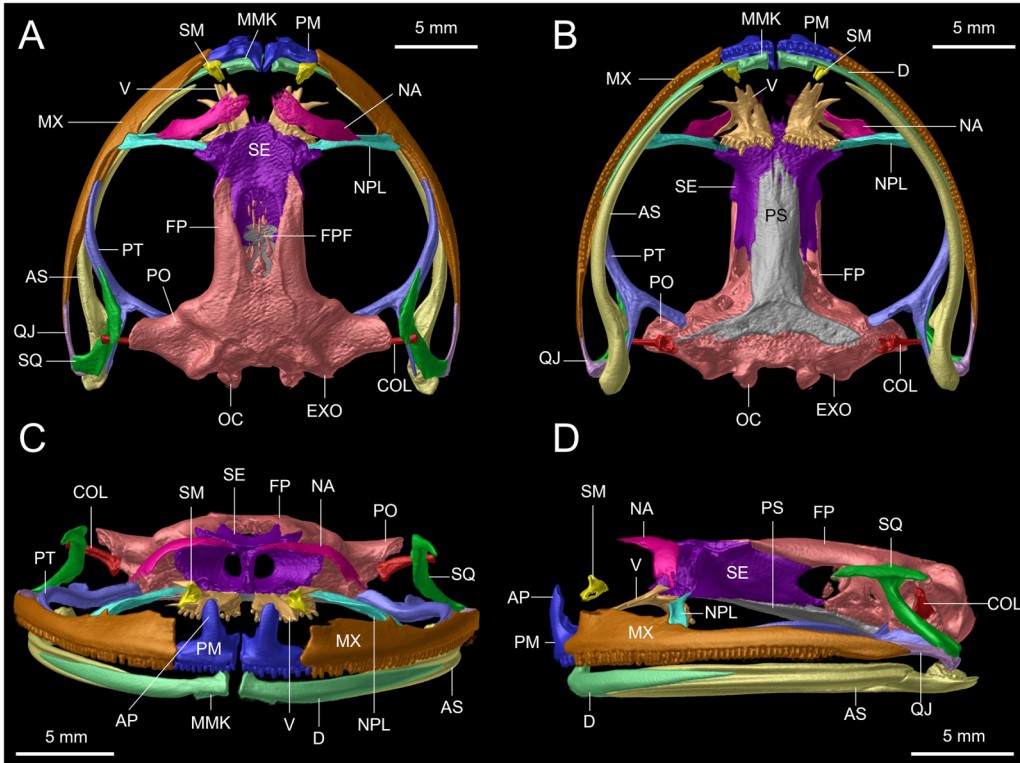

**Figure 12 Osteological details of the cranium of the adult female holotype (DHMECN 14416)** *Hyloscirtus sethmacfarlanei* **sp. nov..** (A) Dorsal view. (B) Ventral view. (C) Anterior view. (D) Lateral view. Labels: AP, alary process of premaxilla; AS, angulosplenial; COL, columella; D, dental; EXO, exoccipital; FP, frontoparietal; FPF, frontoparietal fontanelle; MMK, mentomeckelian bone; MX, maxilla; NA, nasal; NPL, neopalatine; OC, occipital condyle; PM, premaxilla; PO, prootic; PS, parasphenoid; PT, pterygoid; QJ, quadratojugal; SQ, squamosal; SE, sphenethmoid; SM, septomaxilla; V, vomer.

anatomy); in life the dorsal surface with irregular marks mustard yellow heavily stippled with black, especially on flanks and lower back (DHMENC 14549), or a variagated yellow-orange pattern (DHMECN 17554); nictitating membrane dotted with mustard yellow on a gray background (DHMENC 14549) or orange on a black background (DHMECN 17554); extremities orange banded (DHMENC 14549) or spotted (DHMECN 17554), on a grayish black to black ground; flanks black with orange reticulations and irregular spots; throat marbled with irregular orange or yellowish patches with orange tones on a grayish black or black ground; belly and ventral surfaces of the extremities grayish black with irregular sparse diffuse light orange or whitish-yellow patches (DHMENC 14549) or solid orange (DHMENC 17554); palms of hands and feet black with diffuse light orange spots. The uncollected juvenile had a mainly yellow dorsal coloration, with diffuse blackish spots scattered on the flanks and hidden surfaces of the arms and between the fingers, whose tips were yellow. The belly is light cream with diffuse blackish spots. We noted rapid temporal chromatic changes in the juvenile individuals, from dull yellow to orange tones. As observed in other members of the *Hyloscirtus larinopygion* species group, changes in color pattern may be characteristic of different stages of

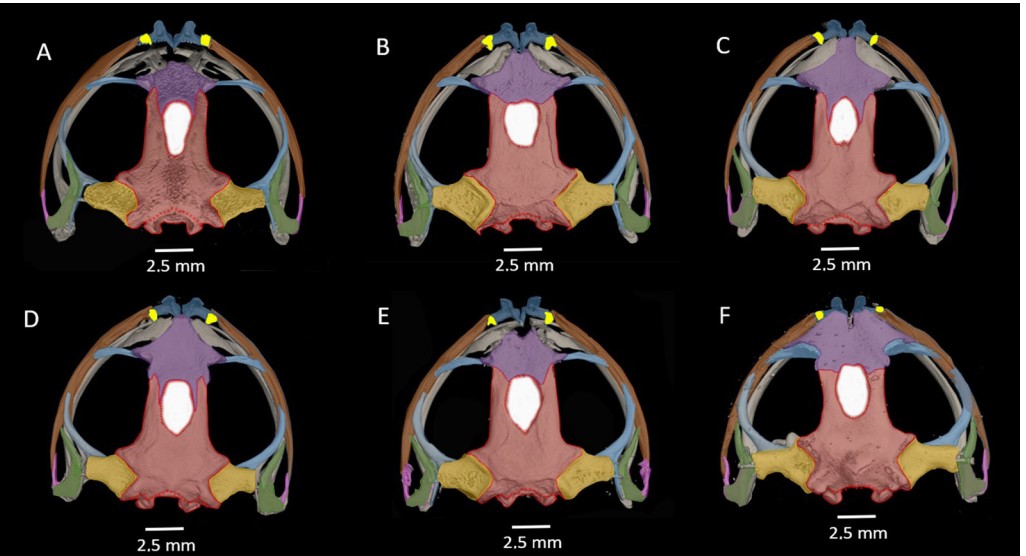

**Figure 13 Dorsal view of CT scan skull detail of *Hyloscirtus sethmacfarlanei* sp. nov. in comparison with other related members of the *H. larinopygion* group.** (A) *Hyloscirtus sethmacfarlanei* sp. nov. (DHMECN 14416). (B) *Hyloscirtus larinopygion* (DHMECN 3799). (C) *Hyloscirtus pacha* (DHMECN 12111). (D) *Hyloscirtus psarolaimus* (DHMECN 6493). (E) *Hyloscirtus lindae* (DHMECN 12483). (F) *Hyloscirtus tapichalaca* (DHMECN 9686). Individual skull diagnostic bones given false colors. Blue: alary process of the premaxilla. Bright yellow: septomaxilla. Brown: maxilla. Purple: sphenoides (in H. tapichalaca this is fused with nasals). Light blue anterior: neopalatine. Light blue posterior: pterigoides. Red anterior: frontoparietals. Posterior red punctuation represents conjunction with exoccipital. White: frontoparietal fontanelle. Light yellow: prootic. Green: squamosal. Pink: quadratojugal.

development and related to ontogenic changes (*Coloma et al., 2012*). The juveniles all shared the same distinctive cloacal ornamentation as the adult (Fig. 10).

**Comparison with similar species (Figs. 8, 9, Table 3).** The black and red pattern of the female of the new species is most similar to the patterns of *Hyloscirtus pantostictus* (*Duellman & Berger, 1982*), from extreme northeastern Ecuador, and *H. princecharlesi Coloma et al. (2012)*, from the Pacific slopes of the Andes of northwestern Ecuador. The new species differs from these in having both the dorsal and ventral surfaces spotted with red (*vs* ventral surface without red spots in *H. pantostictus* and *H. princecharlesi*, Fig. 8), the cloacal ornamentation (Fig. 9) consisting of a well-defined supracloacal fold reaching next to the vent and the presence of a paracloacal fold (*vs* reduced supracloacal fold, without paracloacal fold, not contacting the side of the vent, in *H. pantostictus*, and supracloacal fold defined, reaching the border of the vent, with paracloacal fold thick, in *H. princecharlesi*), and strongly areolate skin texture (*vs* smooth in *H. pacha*, *H. staufferorum*, and *H. larinopygion*). The female of the new species also differs from these two species in having red discs on the tips of all digits (*vs* yellow discs in *H. pantostictus* and grayish discs in *H. princecharlesi*).

The new species' red discs are shared with *H. lindae* (*Duellman & Altig, 1978*) from the eastern Andes, but *H. lindae* does not have red spots on its dorsal surface and does not have a thick supracloacal fold close to the side of the vent (Fig. 9). Juveniles assigned to

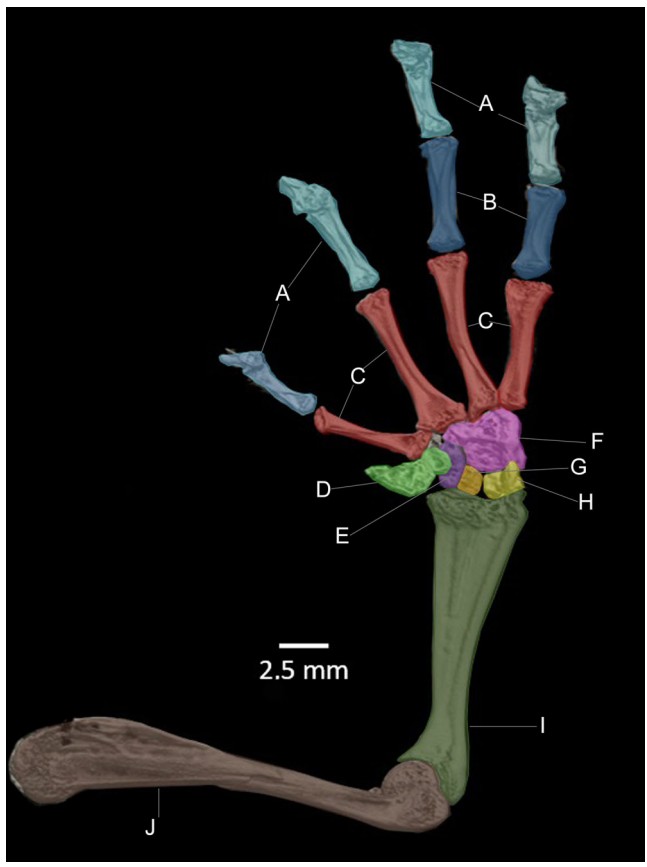

**Figure 14 CT scan of the ventral view of the forelimb bones of the holotype of *Hyloscirtus sethmacfarlanei* sp. nov. (DHMECN 14416).** (A) Distal clawed phalanges (light blue). (B) Medial phalanges (dark blue). (C) Proximal phalanges (red). (D) Prepolex (light green). (E) Distal carpal II (purple). (F) Fused distal Carpals 3+4+5 (pink). (G) Radiale (orange). (H) Ulnare (yellow). (I) Radioulna (dark green). (J) Humerus (brown).

*H. sethmacfarlanei* sp. nov. have a pattern similar to those of *H. princecharlesi* and *H. larinopygion* (*Duellman, 1973*) from northwestern slopes of the Andes. They differ from juveniles of both species in having the dorsum mottled and stippled mustard-yellow and black (*vs* dorsum densely spotted orange-red in *H. princecharlesi*, and yellowish-brown with distinctive cream bars with black interspaces in *H. larinopygion*). The supracloacal fold is well-defined and reaches to the vent in *H. sethmacfarlanei* sp. nov. (*vs* faintly defined and distant from the side of the vent in *H. larinopygion*). *Hyloscirtus sarampiona* (*Ruiz-Carranza & Lynch, 1982*) from the western slopes of the Colombian Andes has dorsal surfaces orange spotted with pale olive, and further differs from the new species by having hidden areas of the limbs, flanks, palmar, plantar surfaces and discs black.

The skull of *H. sethmacfarlanei* sp. nov. (Figs. 12, 13) is generally consistent with those of the other species of the *H. larinopygion* group (*Coloma et al., 2012*). However, there were a few differences between the new species and the species of the group studied by us or by *Coloma et al. (2012)*. In *H. sethmacfarlanei* sp. nov. and *H. ptychodactylus*, the sphenethmoid is not in contact with the nasal, whereas these two bones are in contact in *H. criptico* and in *H. staufferorum*, they are anteriorly in contact in *H. lindae* and
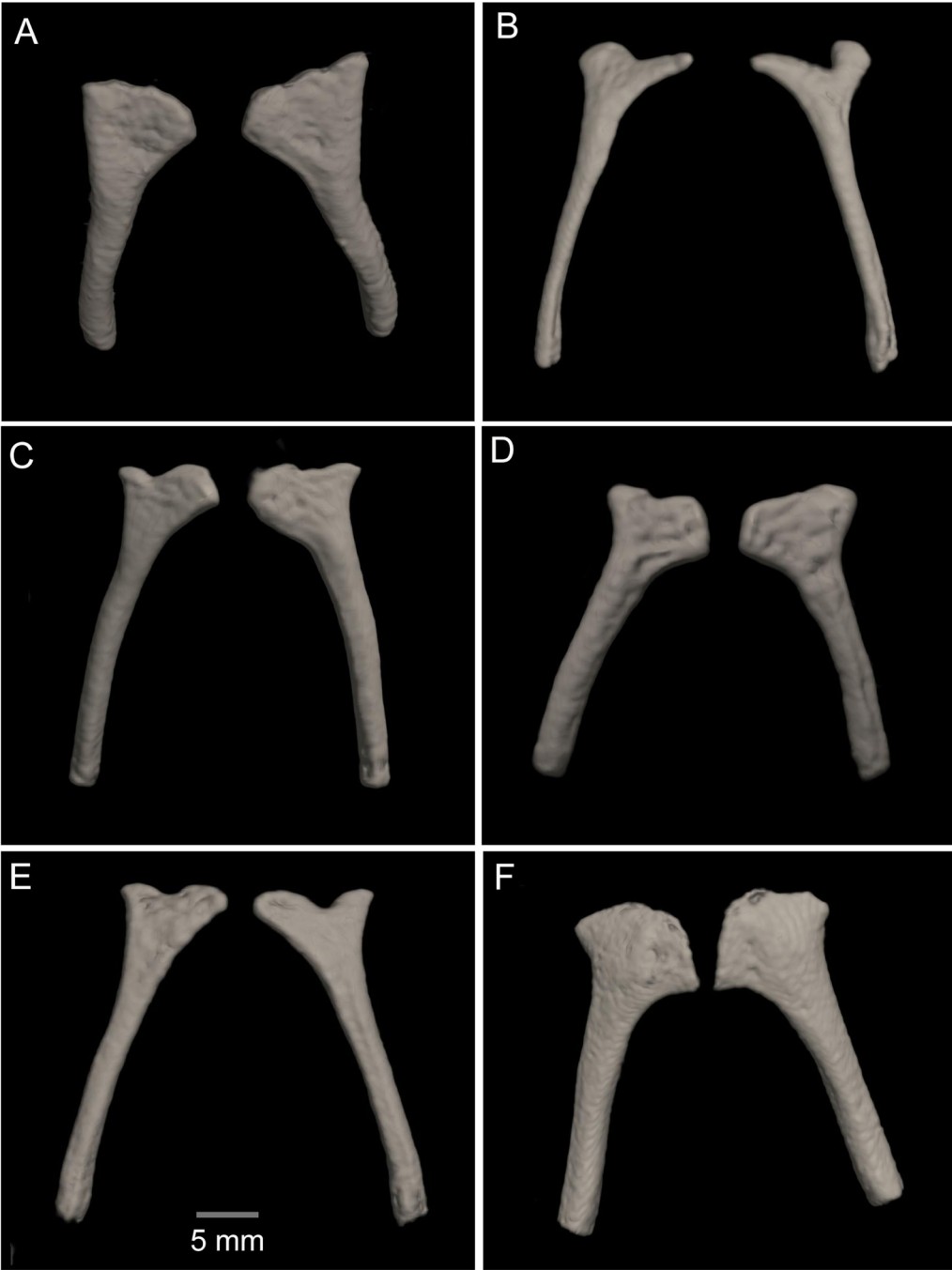

**Figure 15 Comparison of the CT scans of the posteromedials of the hyobranchium in different species of the *Hyloscirtus larinopygion* group.** (A) *Hyloscirtus sethmacfarlanei* sp nov. holotype (DHMECN 14416). (B) *Hyloscirtus larinopygion* (DHMECN 3799). (C) *Hyloscirtus lindae* (DHMECN 12483). (D) *Hyloscirtus psarolaimus* (DHMECN 6493). (E) *Hyloscirtus pacha* (DHMECN 12111). (F) *Hyloscirtus tapichalaca* (DHMECN 9686).

*H. larinopygion*, they are in contact along most of their length in *H. psarolaimus*, they are in contact along their entire length but with still a visible suture in *H. pacha*, and they are completely fused without a visible suture in *H. tapichalaca*. In *H. sethmacfarlanei* sp. nov.,

**Table 3 Cloacal ornamentation and presence of calcar on the heel in Ecuadorian species of the *Hyoscirtus larinopygion* group.**

| Reference and voucher specimens based for cloacal ornamentation | Species | Supracloacal fold | Vent-supracloacal fold relation | Paracloacal fold | Cloacal texture | Calcar on the heel |
|---|---|---|---|---|---|---|
| This work (DHMECN 14416) | *Hyloscirtus. sethmacfarlanei* sp. nov. | Defined | Reaching the vent | Thick, well defined | Strongly, areolate and granular | Present |
| This work (DHMECN 12483) | *Hyloscirtus lindae* | Thick, developed, tongue like shape | Reaching the border of the vent | Weakly defined | Densely tuberculate | Absent |
| This work (DHMECN 12113) | *Hyloscirtus pacha* | Thick | Separated from the vent | Thick, well-defined | Smooth | Present |
| This work (DHMECN 3799) | *Hyloscirtus larinopgygion* | Defined | Separated from the vent | Absent | Smooth | Present |
| This work (DHMECN 6493) | *Hyloscirtus psarolaimus* | Weakly defined | Separated from the vent | Absent | Little tubercles | Absent |
| This work (DHMECN 4590) | *Hyloscirtus criptico* | Defined | Separated from the vent | Weakly defined | Strongly granular | Absent |
| *Duellman & Coloma, 1993* (KU 217695) | *Hyloscirtus staufferorum* | Weakly defined | Separated from the vent | Absent | Smooth | Absent |
| *Duellman & Berger, 1982* (KU190000) | *Hyloscitus pantostictus* | Well-defined | Reaching the border of the vent | Absent | Densely tuberculate | Absent |
| *Coloma et al., 2012* (QCAZ 44893) | *Hyloscirtus princecharlesi* | Defined | Reaching the border of the vent | Present, thick | Smooth | Absent |
| *Mueses-Cisneros & Anganoy-Criollo, 2008* (ICN 53804) | *Hyloscirtus tigrinus* | Thick | Separated from the vent | Present | Smooth with granular tubercles | Present |
| *Duellman & Hillis, 1990* (KU 209780) | *Hyloscirtus ptychodactylus* | Defined | Reaching the border of the vent | Present | Smooth | Present |
| *Kizirian, Coloma & Paredes-Recalde, 2003* (QCAZ 65235) | *Hyloscirtus tapichalaca* | Defined | Reaching the border of the vent | Present, thick | Smooth | Absent |
| *Almendáriz et al., 2014* (QCAZ 68646 ) | *Hyloscirtus condor* | Thick | Reaching the border of the vent | Absent | Smooth | Present |
| *Ron et al., 2018* (QCAZ 40331) | *Hyloscirtus hillisi* | Well-defined | Separated from the vent | Present, thin | Areolate | Absent |

*H. lindae*, *H. pacha*, and *H. larinopygion* the nasal is not in contact with the maxilla, whereas it is in contact with the maxilla in *H. criptico*, *H. pantostictus*, *H. ptychodactylus*, *H. staufferorum*, and *H. tapichalaca*. The frontoparietals of *H. sethmacfarlanei* sp. nov. are comparatively more rugose than in other species of the group (Fig. 12). In contrast to *H. pantostictus* and *H. staufferorum*, in *H. sethmacfarlanei* sp. nov., *H. criptico*, *H. lindae*, *H. pacha*, *H. psarolaimus*, *H. ptychodactylus*, *H. larinopygion*, and *H. tapichalaca* the otic ramus of the squamosal is not in contact with the prootic. In *H. sethmacfarlanei* sp. nov.

the zygomatic ramus of the squamosal is only slightly longer than otic ramus, whereas it is moderately longer than the otic ramus in *H. pacha* and *H. staufferorum*, and distinctly longer than the otic ramus in *H. criptico, H. lindae, H. pantostictus, H. ptychodactylus, H. larinopygion*, and *H. tapichalaca*. In contrast to *H. criptico, H. larinopygion*, and *H. tapichalaca*, in *H. sethmacfarlanei* sp. nov., *H. lindae, H. pantostictus, H. ptychodactylus*, and *H. staufferorum* the vomers are not in medial contact. *Hyloscirtus sethmacfarlanei* sp. nov. has 12-13 vomerine tooth loci, 54-56 tooth loci on each maxilla, and 10-11 tooth loci on each premaxilla, whereas we counted 14 vomerian tooth loci, 59-60 maxillary tooth loci, and 11-12 premaxillary tooth loci in *H. lindae*, 14 vomerine tooth loci, 52-59 maxillary tooth loci, and 9 premaxillary tooth loci in *H. pacha*, 13-14 vomerine tooth loci, 52-53 maxillary tooth loci, and 9 premaxillary tooth loci in *H. psarolaimus*, 11-12 vomerine tooth loci, 56 maxillary tooth loci, and 12 premaxillary tooth loci in *H. larinopygion*, and only 5-6 vomerine tooth loci, 31-33 maxillary tooth loci, and 5-6 premaxillary tooth loci in *H. tapichalaca*.

In the new species the posteromedial processes of the hyobranchium possess a triangular shaped anterior portion, and a shorter posterior portion compared with the other species shown in Fig. 14, which have an external round border and an internal spine-like border. In *H. lindae, H. psarolaimus* and *H. pacha*, the anterior portions have rounded external and internal borders. In *H. tapichalaca* it is broad and "shell-like" in its anterior border.

There are no relevant differences between the forelimb bones of the new species and those of the other species in the group, with the exception of male specimens of *H. tapichalaca*, which have a greatly enlarged prepollex (*Kizirian, Coloma & Paredes-Recalde, 2003*; this study) compared to the other species of the *H. larinopygion* group.

**Distribution (Fig. 16).** *Hyloscirtus sethmacfarlanei* sp. nov. is known at the moment only from the type locality in Fundación EcoMinga's Machay Reserve, Cerro Mayordomo, 2,970 m altitude, in the eastern cordillera of the central Ecuadorian Andes, in the northern side of the upper Rio Pastaza watershed near the southern border of Llanganates National Park in the province of Tungurahua.

**Natural history.** The type locality consists of dwarf open mossy forest, covered with bryophytes and epiphytes, and saturated with humidity. All four known individuals of this species were found on a single narrow mountain ridge, in bromeliads of the genus *Guzmania* growing within 60–90 cm above the ground (Fig. 17). The holotype is an adult gravid female with a mass of eggs in early stage of development in March 2018. Adult male, tadpole and advertisement call remain unknown.

There is some evidence that the striking coloration of the adult female of *H. sethmacfarlanei* sp. nov. could be aposematic. The frog's discoverer (Darwin Recalde), after briefly handling the frog, noticed an unpleasant tingling sensation down his arm, not restricted to the area which had contacted the frog; the sensation lasted several hours. Fausto Recalde, who had shorter contact with the frog, developed similar but

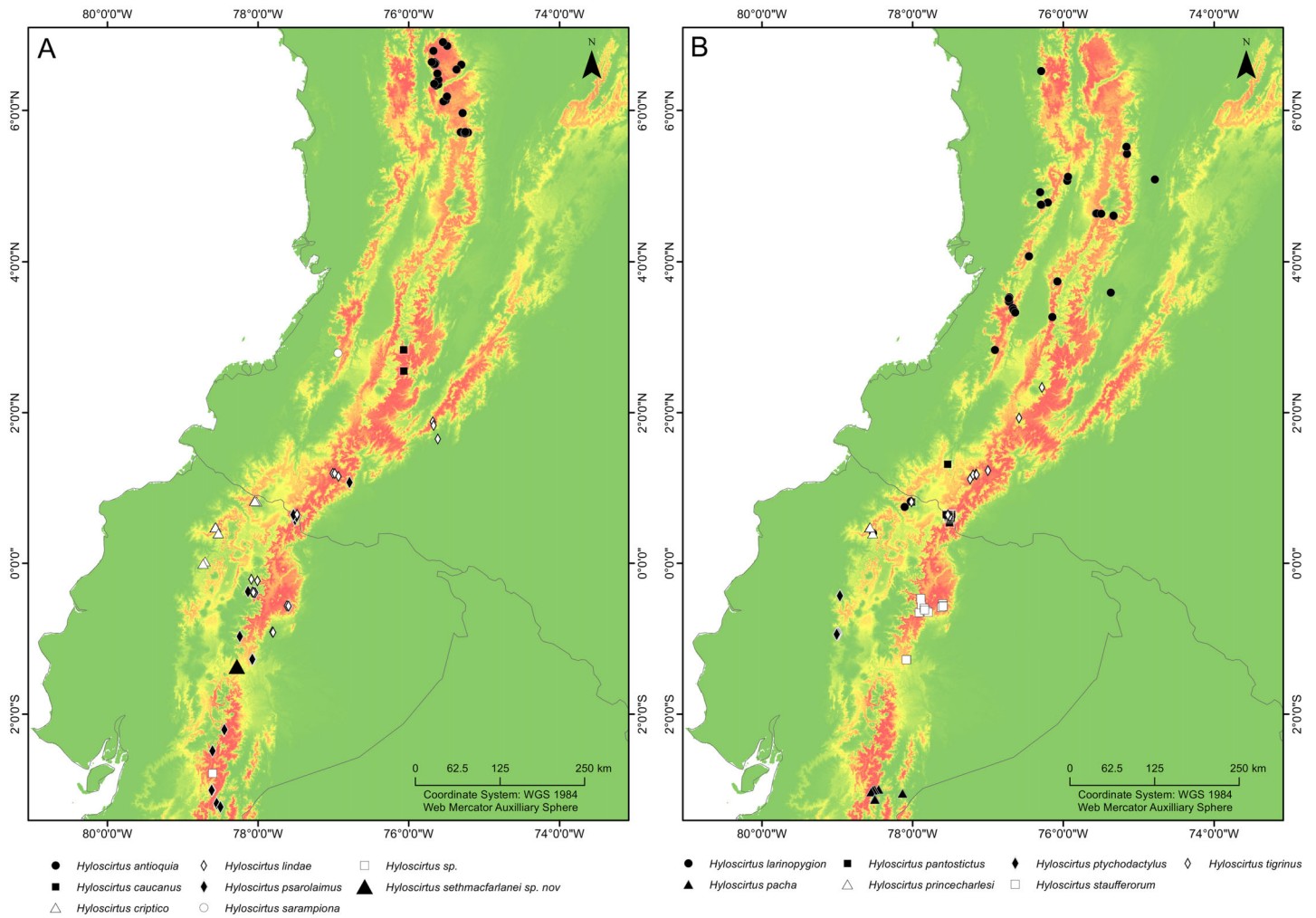

**Figure 16** **Maps of northwestern South America showing the ecological niche modeling for all species of the northern clade of the *Hyloscirtus larinopygion* species group (yellow to red shadows).** (A, B) Both maps show the same ecological niche model but species were divided into two maps to allow locality points for all species to be included. Type locality of *H. sethmacfarlanei* sp. nov. indicated by a black triangle in (A).

shorter-lasting symptoms; these reactions were not observed after handling the juveniles. During handling of the holotype specimen in the museum, it emitted a white exudation on dorsal surfaces with a distinctive odor similar to diluted contact cement. Additionally, when tissue was taken from the liver, dark blackish-colored blood was observed.

The bright yellow uncollected juvenile slept during the day, and when disturbed, it adopted a defensive ball-like position, as observed in other species of the *H. larinopygion* group (*Kizirian, Coloma & Paredes-Recalde, 2003*; *Bejarano-Muñoz, Perez Lara & Brito Molina, 2015*). Thus the juvenile coloration may also advertise distastefulness.

Nocturnal surveys done by our team in the habitat of *Hyloscirtus sethmacfarlanei* sp. nov. revealed three sympatric anuran species: two undescribed *Pristimantis* species and one species of the *Pristimantis buckleyi* complex.

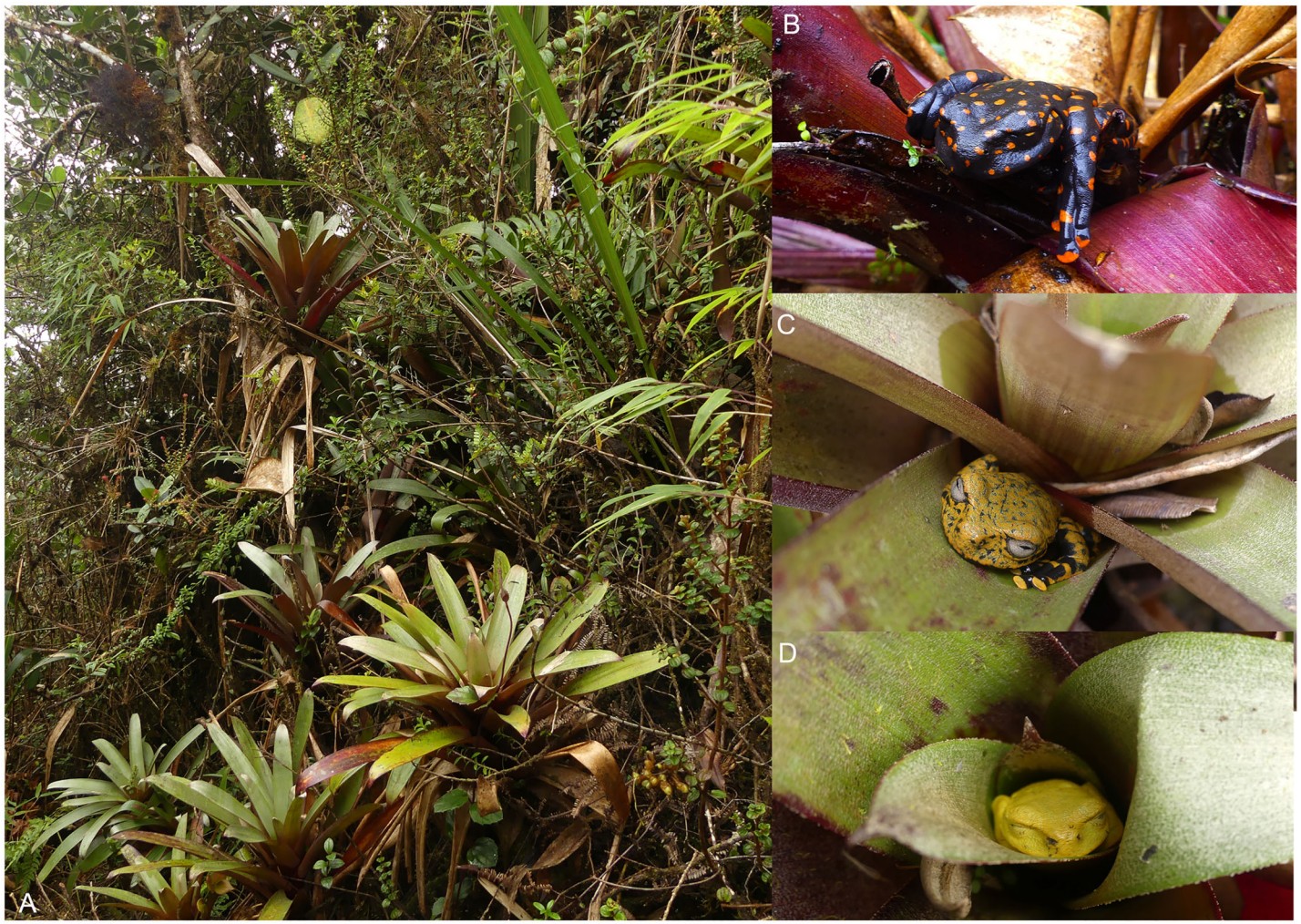

**Figure 17** *Hyloscirtus sethmacfarlanei* **sp. nov., at the type locality and its habitat.** (A) Humid cloud forest with *Guzmania sp.* bromeliads at type locality. (B) Female Holotype (DHMECN 14416). (C) Juvenile paratype (DHMECN 14549). (D) Uncollected juvenile. Photographs: FR, JPRP.

**Conservation Status.** All four known individuals are known only from the same few square meters of ridgeline, but the area is poorly studied and inaccessible because of steep topography. We suggest the IUCN category Data Deficient (DD) for this species.

**Etymology.** The specific epithet *sethmacfarlanei* is a patronym in honor of Seth MacFarlane, American writer, director, producer, actor, artist, musician and conservationist, with an outstanding passion for science, biodiversity and the natural world.

## DISCUSSION

Despite the low number of studied specimens, limited to the type material for the new species, the congruence of strong molecular and morphological evidence supports the hypothesis that all these specimens belong to a new monophyletic taxon in the *Hyloscirtus larinopygion* species group (Fig. 1), which has been evolving independently from other

named taxa for many millions of years (see below), meeting the core criterion for species recognition (see *Simpson, 1951*, *1961*; *Wiley, 1978*; *De Queiroz, 2007*). The collected juveniles and also the small uncollected young shared the same cloacal ornamentation and skin texture as the adult female, and the performed genetic analyses confirmed that the adult female and the single sequenced juvenile belong to the same species.

Though only one adult female specimen is known, we do not expect this color to vary much among other female individuals of the species, based on the lack of significant variation across individual adult female members of each of the other species in this group (*Coloma et al., 2012*). Some of the coloration differences between the female and the three observed juveniles may be related to ontogenetic changes, with larger probably subadult individuals displaying color patterns more similar to the described female pattern, as seen in other species of the *Hyloscirtus larinopygion group* (*Coloma et al., 2012*). Additionally field and laboratory observations on live juvenile specimens show slight differences of coloration in the same specimen under different background or conditions, ranging from yellow to orange tones. The color change in animals can be adaptive phenotypic plasticity in heterogeneous environments (*Kang, Kim & Jang, 2016*).

The *Hyloscirtus larinopygion* group is characterized by overlapping morphological and morphometric characters. In many cases, the preserved and living coloration patterns continue to be the main trait used to discriminate externally between the species in this group (*Duellman & Hillis, 1990*; *Duellman & Coloma, 1993*; *Coloma et al., 2012*; *Rivera-Correa & Faivovich, 2013*; *Rivera-Correa, García-Burneo & Grant, 2016*; *Ron et al., 2018*). Our analyses of micro-CT scan osteology, skin texture, and supracloacal folds show the importance of continuing to incorporate and explore additional evidence to help delimit lineages of the group, whose evolutionary radiation in the Ecuadorian Andes is apparently still underestimated.

Most osteological characters do not seem to vary greatly among the different species of the *H. larinopygion* group. Even though we were able to detect some differences that could be of diagnostic value, we only had the opportunity to osteologically examine one individual from each of the six species. In previous studies (*Kizirian, Coloma & Paredes-Recalde, 2003*; *Coloma et al., 2012*), another 15 specimens from eight species could be examined, so that data for osteological comparisons are available for a total of eleven species, comprising three specimens each from four species, two specimens each from two species and only one specimen each from five species. Some of the differences found between species might be less clear with a larger sample size. In *H. pacha*, for example, the vomers are in medial contact in the two specimens studied by *Coloma et al. (2012)*, while they were not in contact in our individual. The opposite is true for *H. psarolaimus*, where the vomers are not in contact in the individual studied by *Coloma et al. (2012)* and are in contact in our individual (Fig. 13). Furthermore, in Coloma's individual of *H. psarolaimus*, the nasal and maxilla are in contact, and the zygomatic ramus of the squamosal is approximately as long as the otic ramus. In our individual, however, the nasal and maxilla are not in contact and the zygomatic ramus is moderately longer than the otic ramus (Fig. 13). On the other hand, we could not detect any osteological difference between the individual of *H. lindae* that we examined and the two specimens of that species examined

by *Coloma et al. (2012)*, nor between the individual of *H. tapichalaca* we examined and the two specimens of that species examined by *Kizirian, Coloma & Paredes-Recalde (2003)*. Fortunately, modern non-invasive techniques such as micro-CT scanning are now increasingly available to quickly visualize the skeletal anatomy of a specimen in three dimensions. Since dissection is not involved, multiple specimens of a species can be easily scanned and compared. In the future, many more individuals of the various species of the *H. larinopygion* group will hopefully be studied using this technique, so that we can get a more accurate picture of the osteological differences between the various species.

Biogeographic interpretations of the evolutionary history of *H. sethmacfarlanei* sp. nov. would be too speculative, mainly because the sister relationship between the new species and other *Hyloscirtus* has low bootstrap support (Fig. 1). Our inferred phylogeny recovered two species (*H. armatus* and *H. charazani*) of the *H. armatus* group as part of the *H. larinopygion* (Fig. 1), but, again, with low bootstrap support. Other recent studies (*Coloma et al., 2012*; *Ron et al., 2018*) have found strong support for the monophyly of the *larinopygion* and *armatus* groups. Therefore, differences might be a consequence of different gene and taxon sampling schemes.

The fossil-calibrated divergence times between some of the species in the *H. larinopygion* group were estimated by *Coloma et al. (2012)*, presented in their Fig. 5. The estimated divergence times range from 32 Mya +/− 12 My between *H. tapichalaca* and its nearest relatives, to 2.6 Mya +/− 1 My between *H. pacha* and *H. staufferorum*. We have used genetic distances to fit *H. sethmacfarlanei* into the chronology proposed in this tree. If the calibrated phylogenies of *Coloma et al. (2012)* are correct, our analysis suggests that the divergence between *H. sethmacfarlanei* and the other known species of the group preceded the Quaternary period. The calculation of genetic distance does not use information about the topology of the phylogenetic tree, so it does not depend on the chosen topology and is not affected by the degree of bootstrap support for the most likely tree.

The shortest genetic distances between *H. sethmacfarlanei* p. nov. and its relatives (2.2–2.9%) are considerably greater than the genetic distances between some other clearly-defined species in the *H. larinopygion* group, such as the distance between *H. ptychodactylus* and *H. princecharlesi* (1.3%). The distances between our species and the other species in the group are also considerably greater than the observed differences between individuals within a species (Table 1). Thus our taxonomic proposal is consistent (in terms of genetic distance and divergence times) with past taxonomic decisions in this group.

The two sequenced specimens of *H. sethmacfarlanei* sp. nov. show a genetic distance of 0.4%, although they come from exactly the same location. This degree of divergence within a population is about average for sequenced conspecific members of the species in the *H. larinopygion* group (0.2–0.9%; *Coloma et al., 2012*). With only two sequenced specimens, our conclusions from this are necessarily limited, but increased sampling can only increase the percentage of polymorphic single-nucelotide loci detected in the population. The observed level of heterozygosity between two randomly selected individuals would not be possible if the population were highly inbred, implying that the

actual population is not exceptionally small (*Jetz & Pyron, 2018*; *Lyra et al., 2020*). The forest at the type locality of the new species, at 2,900–3,000 m on Cerro Mayordomo, is continuous with similar forest on the Cerro Hermoso massif in the center of Los Llanganates National Park, 17 km to the north of the type locality. The new species probably occupies at least this range, all of it consisting of undisturbed forest. During the Holocene glacial maximum this forest community would probably have moved down the mountains by 1,000 m (*Dodson, 2003*), potentially connecting this population to many other nearby mountains.

Ecological niche modeling is a powerful tool for biogeographic analyses. Bioclimatic modeling approaches have been applied beyond single species distribution models to identify the potential distribution of undiscovered taxa, understand the ecological niche of supra-specific taxa, or predict the community structure of multiple species assemblages (*e.g.*, *Larsen et al., 2012*; *Ihlow et al., 2016*; *Braun et al., 2019*). A Maxent model was applied to the known species of the *H. larinopygion* clade to estimate its potential distribution (though it does not take into account the history of past connectivity between sites). Modeling the distribution of supra-specific taxa assumes that members of the taxon respond similarly to environmental conditions. This approach is considered appropriate for the northern clade of *H. larinopygion* group due to their occurrence in apparently similar ecosystems and habitats across their distribution (*Duellman & Hillis, 1990*; *Kizirian, Coloma & Paredes-Recalde, 2003*; *Coloma et al., 2012*; *Almendáriz et al., 2014*; *Rivera-Correa, García-Burneo & Grant, 2016*; *Ron et al., 2018*; *Ron, Merino-Viteri & Ortiz, 2021*).

It is remarkable that despite intensive research work in the upper Rio Pastaza watershed and in the *Hyloscirtus larinopygion* species group, researchers still continue to discover conspicuous new species in the group. Our Maxent model estimates the potential distribution of all members of the clade, showing areas where potential undiscovered species might occur. The Maxent model shows that the type locality of *H. sethmacfarlanei* sp. nov. is within the predicted range of niches of the northern clade of the *H. larinopygion* group. Many additional areas across the Andes of Colombia and Ecuador show high probability of occupation according to the model, but no species records, *e.g.*, the Cordillera Oriental of Colombia, the southern Cordillera Occidental of Colombia, and the extreme northern and central Cordillera Oriental of Ecuador (Fig. 15).

## CONCLUSIONS

We present converging lines of evidence supporting our conclusion that a newly discovered population of *Hyloscirtus*, belonging to the *H. larinopygion* group, represents a distinctive new species. Our observations on its anti-predatory behavior lead us to conclude that this species is almost certainly toxic and/or unpalatable, and that its bright colors are probably aposematic. Our genetic analysis suggests that *Hyloscirtus sethmacfarlanei* sp. nov. is an older species, not a product of Quaternary isolating mechanisms. Our study further confirms the importance of the Llanganates – Sangay Ecological Corridor, outside of Ecuador's national park system, as a center of endemism and diversity. Additionally, a distribution model for the *H. larinopygion* species group

suggests many other potential areas of occurrence along the northern Andes for members of this group.

## APPENDIX 1

List of specimens examined of the *Hyloscirtus larinopygion* group (13). *Hyloscirtus criptico* (1): Ecuador: Carchi, Morán, DHMECN 15831; *H. larinopygion* (1): Ecuador: Carchi, Morán, DHMECN 3799; *H. lindae* (1): Ecuador: Napo, Guango Lodge, DHMECN 12483; *H. pacha* (4): Ecuador: Morona Santiago, Guabisai, DHMECN 12110–12113; *H. psarolaimus* (4): Ecuador: Sucumbíos, La Bonita, DHMECN 6493–6496; Morona Santiago; Zuñac, DHMECN 12114; *H. sarampiona* (1): Colombia: Valle del Cauca, Quebrada Sopladero, Holotipo ICN 7440; *H. tapichalaca* (1): Ecuador: Zamora Chinchipe, Reserva Tapichalaca, DHMECN 9686.

## ACKNOWLEDGEMENTS

Special thanks to the Rainforest Trust and World Land Trust for supporting Fundacion EcoMinga's efforts to protect the forests of the upper Rio Pastaza watershed and WWF Ecuador. Thanks to Fundación Ecominga and its supporting staff: Javier Robayo, Santiago Recalde, Jesus Recalde, Jordy Salazar, Piedad Paredes. We thank Diego Inclán and Miguel Urguiles Merchán from INABIO, and Yaneth Muñoz and Juan C. Sánchez of the ICN, for the facilities provided for the examination of the type material in their charge. We also thank Santiago Ron and Jorge Brito for use of photographic records.

### Funding

This work was supported by the Inédita Program of the Ecuadorian Science Agency SENESCYT (Respuestas a la Crisis de Biodiversidad: La Descripción de Especies como Herramienta de Conservación; INEDITA PIC-20-INE-USFQ-001) and grants from John Moore to the Population Biology Foundation. The funders had no role in study design, data collection and analysis, decision to publish, or preparation of the manuscript.

### Grant Disclosures

The following grant information was disclosed by the authors:
Inédita Program of the Ecuadorian Science Agency SENESCYT (Respuestas a la Crisis de Biodiversidad: La Descripción de Especies como Herramienta de Conservación; INEDITA PIC-20-INE-USFQ-001).

### Competing Interests

The authors declare that they have no competing interests.

### Author Contributions

- Juan P. Reyes-Puig conceived and designed the experiments, performed the experiments, analyzed the data, prepared figures and/or tables, and approved the final draft.

- Darwin Recalde conceived and designed the experiments, prepared figures and/or tables, and approved the final draft.
- Fausto Recalde performed the experiments, prepared figures and/or tables, and approved the final draft.
- Claudia Koch analyzed the data, authored or reviewed drafts of the article, and approved the final draft.
- Juan M. Guayasamin performed the experiments, analyzed the data, authored or reviewed drafts of the article, and approved the final draft.
- Diego F. Cisneros-Heredia performed the experiments, analyzed the data, authored or reviewed drafts of the article, and approved the final draft.
- Lou Jost conceived and designed the experiments, authored or reviewed drafts of the article, and approved the final draft.
- Mario H. Yánez-Muñoz conceived and designed the experiments, performed the experiments, analyzed the data, prepared figures and/or tables, and approved the final draft.

### Animal Ethics

The following information was supplied relating to ethical approvals (*i.e.*, approving body and any reference numbers):

This study was approved by the Ministerio del Ambiente Republica del Ecuador to Instituto Nacional de Biodiversidad(INABIO)-MAE-DNB-CM-2016-0045 and MAE-DNB-CM-2019-0120).

### Field Study Permissions

The following information was supplied relating to field study approvals (*i.e.*, approving body and any reference numbers):

The field permit was issued by the Ministerio de Ambiente Republica del Ecuador (MAE-DNB-CM-2016-0045 and MAE- DNB-CM-2019-0120).

### DNA Deposition

The following information was supplied regarding the deposition of DNA sequences:

The sequences of the new species are available in GenBank:

Hyloscirtus_sp_DHMECN_14416: OM293945

Hyloscirtus_sp_DHMECN_14549: OM293946

### Data Availability

The CT scan data are available at MorphoSource:

https://doi.org/10.17602/M2/M458232

https://doi.org/10.17602/M2/M458252

https://doi.org/10.17602/M2/M458264

https://doi.org/10.17602/M2/M458273

https://doi.org/10.17602/M2/M458289

https://doi.org/10.17602/M2/M458297
### New Species Registration

The following information was supplied regarding the registration of a newly described species:

Publication LSID: urn:lsid:zoobank.org:pub:4BF8C735-F06C-41AE-B130-EE41130535CC

Hyloscirtus sethmacfarlanei LSID: urn:lsid:zoobank.org:act:4664187C-9C99-443B-8831-5AD93E8A817A

### Supplemental Information

Supplemental information for this article can be found online at http://dx.doi.org/10.7717/peerj.14066#supplemental-information.

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
