# Peer review of "A spectacular new species of Hyloscirtus (Anura: Hylidae) from the Cordillera de Los Llanganates in the eastern Andes of Ecuador"

_PeerJ, doi:10.7717/peerj.14066_

## Round 0.1 · original submission · Major Revisions

Dear authors,

We have received three review reports for your manuscript entitled “A spectacular new species of Hyloscirtus (Anura: Hylidae) from the Cordillera de Los Llanganates in the eastern Andes of Ecuador”. They include important suggestions and recommendations that I would like that you take in consideration to improve your work. Two of the reviewers also provided an annotated pdf file with a more detailed revision of the manuscript that will be useful to identify the sections that will need the major attention and changes.

I have also prepared an annotated pdf file, which will be sent to you along to this letter.

As you will see from the reports, there are at least three delimited areas that present the main problems, and within which you will have to work with effort to improve them:

1-The erection of a new species based on just one or perhaps two specimens (of which just one is assumed to be an adult) in an extant group of vertebrates. Here, the main problem is allied to the implications of dorsal pattern and body color to characterize the new species, conditions that are different in the studied specimens.

2- The lack of a deep analysis of the resulting phylogeny. The topology based on mitochondrial data should be compared to those previously obtained from morphologic and morphometric characters.

3- Figures and references. Figures include nice photographs of the specimens but are poorly informative in their captions; even some errors were detected which are all specified in the annotated pdfs. Regarding the “References” section, it seems that some papers are missing and have to be added.

Point 1 is one important, if not the most important aspect that must be improved. For such purpose, I have marked in the annotated pdf some features that have to be addressed. The most important is the issue generated around the changes in dorsal pattern and body color that are very frequent in these frogs. You give minimal attention to the fact that you are describing two different dorsal patterning against a possible similar color background that you are attributing to ontogeny but you do not mention it. It is interesting because in general, the changes follow similar dorsal pattering against the different color background that differentiate juveniles from adults. Therefore, I would like to know how you identify one of the specimens as an adult and why the other as a juvenile. The same is valid for the determination of the sexual attributions that you suggested something that is not easy to do in frogs, especially in juveniles. You would have also to include a section in the Discussion that will be exclusive for analyzing different dorsal patterns and body color that you found in the two described specimens, including diverse factors that could produce the change: environmental (also if related to rapid changes in the environmental conditions, particularly of temperature as was previously suggested), physiological or if they respond to ontogenetic changes, and if they can be related to sexual differences. The inclusion of this section will generate new references that you will have to add to the References section.

Figures will have to be improved following the commentaries added to the annotated pdfs from reviewers and from myself.

Lastly but not least I recommend that you leave the conclusions as conditional to the finding of more specimens that can be recognized as belonging to this species, in order that the corroboration of the hypotheses suggested in this manuscript can be guaranteed.

I hope that you find useful the recommendations kindly provided by the reviewers and those included in this letter, to revise your manuscript and submit a new improved version soon. Otherwise, you can always provide arguments supporting a different point of view in the rebuttal letter to the editor.

With my best wishes,
Graciela Piñeiro

·

Basic reporting

I have carefully read the manuscript entitled "A spectacular new species of Hyloscirtus (Anura: Hylidae) from the Cordillera de Los Llanganates in the eastern Andes of Ecuador (#69046)" by Reyes-Puig et al. In that work the authors describe a new species of Hyloscirtus based on morphological, osteological and molecular evidence (also niche models support the authors' conclusions).

I consider that the manuscript is very well written, it is clear and the evidence presented supports the assignment of the specimens collected in Cerro Mayordomo (Tungurahua Province, Ecuador) to a new taxon. The text is clear, concise and concrete. I'm not a native English speaker, so I can't judge if it's grammatically correct, but I had no difficulty reading and understanding everything the authors wrote. The figures look superb (specially specimens photos and illustrations of hand and foot of the holotype of Hyloscirtus sethmacfarlanei), and the literature is up-to-date and pertinent.

The most important question I have is the validity of a description of a new taxon based on a single adult specimen (the authors use two specimens but one of them is a juvenile). On the other hand, I think that it is not convenient to use a juvenile specimen as type material (perhaps the juvenile collected could be "other complementary material", and wait for a finding of male specimens).

Experimental design

The manuscript deals with the description of a new species. In this sense there is no "experimental design" stricto sensu, however the methodology used for the description is in accordance with the standards of recent descriptions. The approach is multiple since the authors gather morphological (external characteristics, coloration) and anatomical (osteology) evidence. To this they add support based on molecular data, and niche models. All the evidence is consistent and supports that the population of Hyloscirtus found constitutes an independent lineage that justifies its description as a new taxon.

Some items need to be corrected / explained:

See line 145, please tell us how you get 8 sequences from two specimens (holotye and paratype).

There is an error in Figure 11. The figure caption repeats the text of Figure 12 referring to the forelimb bones; however that figure contains osteological details of the crania of different species of Hyloscirtus.

Validity of the findings

The validity of the findings could be compromised by the low number of specimens used in the description. This is a weakness in the entire work (for example, morphological and osteological characters are compared on the basis of a single individual).

Additional comments

See attached file

Reviewer 2 ·

Basic reporting

In general terms, the manuscript is well structured, accompanied by images that support its results.

Experimental design

The research is ethical with methods according to the requirements of the topics covered in the research.

Validity of the findings

There is evidence for the proposed taxonomic decision. In particular the phylogenetic position of the new entity. However, there are still weaknesses in the delimitation from a morphological perspective

Additional comments

I have made important comments in the hope that they will be taken into account to improve the manuscript version. Avoid personal assessments, as this is a scientific product. The section that requires the most work and is the most sensitive in taxonomy is the comparative diagnosis, especially in a group where coloration has been, by inertia, the most traditional form of delimitation.

The authors have sufficient taxonomic experience and I am convinced that they will explore a new set of characters and will be able to do so with ease.

In addition, the discussion should be taken up again, especially since they had difficulty understanding the inferred phylogenetic relationships presented in Figure 1.

Annotated reviews are not available for download in order to protect the identity of reviewers who chose to remain anonymous.

·

Basic reporting

Dear Editor,
This manuscript describes a new species of Hyloscirtus from the Andes of Ecuador. The evidence to demonstrate that the species is undescribed is convincing. The manuscript is well organized and is written in clear, unambiguous, professional English. A few suggestions of style are made below.

Abstract: The first sentence comprises most of the abstract, it is long and, therefore, difficult to understand. Please consider splitting it into smaller units. In addition, the abstract should provide more information. I suggest adding results on the phylogenetic relationships of the new species.

The Introduction provides an adequate review of the relevant literature. Information is correctly referenced but a few exceptions are listed below:
• Lines 64-65. “The genus is characterized mainly by…” Please provide a reference for this statement.
• Lines 65-66. “At the time of the last revision, all known species were thought to reproduce alongside rushing streams” Also needs a reference.

The article is well organized and conforms to the journal standards.

Figures are relevant and of good quality. One suggestion regarding the CT-scan figures (e.g., Figs. 10, 11): it looks like they have been manipulated to highlight differences among bones. Those changes are fine, but they should be explained in the Methodology section or in the Figure legends.
The legend of Figure 11 does not correspond to the figure, please check.

Minor comments

• Line 64. Replace “a well developed lateral membrane” for “well-developed lateral fringes”. The correct terminology for the referred structure is “fringe”, see, for example, Faivovich et al. (2005).

• Line 103. Style suggestion. Change “day walk” by “diurnal walk”.

• Line 142. “Coloma et al. (2002)” Either the authors are referring Coloma et al. (2012) or the reference is missing in the Literature Cited.

• Line 164. The species name should be in italics.

• Line 192. “sp. nov.” should not be in italics

• Line 276-277. “toes long, narrower than the hand” The authors probably mean “toes long, narrower than the disk”?

• Line 281. “Body skin” probably refers to “Skin on dorsum”

• Line 292. “tips of digits are marked with white spots” This description does not agree with Figure 2, please check.

• Lines 304-309. Consider changing the word “ground” for “background”

• Line 343. “2,972 m” change to “2972 m” For consistency, avoid using the comma as a decimal separator.

• Line 383. “H.” should be in italics

• Line 504. The species name should be in italics.

Experimental design

The manuscript falls within the aims and scope of the journal.
The research question is well defined, relevant, and meaningful, as species descriptions usually are.
The methods are described with enough detail. My only minor request is to explain what was the outgroup and how was the tree rooted.

Validity of the findings

The authors should be commended by providing convincing evidence of the validity of the species being described. The genetic evidence is adequate, and the morphological distinctiveness of the examined individuals indicate that they belong to a previously undescribed species.
My main suggestion regarding the analyses is to delete the ecological niche modelling section. I did not find in the text a justification to carry it out and its findings are not presented in the Results section. The niche modelling seems out of place because it does not provide evidence to support the uniqueness of the new species. In addition, environmental niche modelling is carried out for individual species. However, the authors are applying it to a group of species which is not a standard practice.

The authors should compare their phylogeny to previous published phylogenies of Hyloscirtus. It is a bit odd that in their tree, the H. larinopygion species group is not monophyletic. Most previous phylogenies have recovered it as a monophyletic group (e.g., Lyra et al. 2020, Jetz and Pyron 2018). The authors should comment on that difference.

Line 359: Change “is aposematic” to “is likely aposematic” The hypothesis of aposematism is reasonable but needs to be corroborated with more than anecdotical evidence.

Literature cited:

Faivovich J, Haddad CFB, Garcia PCA, Frost DR, Campbell JA, Wheeler WC (2005) Systematic review of the frog family Hylidae, with special reference to Hylinae: Phylogenetic analysis and taxonomic revision. Bulletin of the American Museum of Natural History 294: 1–227.

Jetz W, Pyron RA (2018) The interplay of past diversification and evolutionary isolation with present imperilment across the amphibian tree of life. https://doi.org/10.5061/dryad.cc3n6j5.Phylogeny

Lyra ML, Lourenço ACC, Pinheiro PDP, Pezzuti TL, Baêta D, Barlow A, Hofreiter M, Pombal JP, Haddad CFB, Faivovich J (2020) High-throughput DNA sequencing of museum specimens sheds light on the long-missing species of the Bokermannohyla claresignata group (Anura: Hylidae: Cophomantini). Zoological Journal of the Linnean Society 190: 1235–1255. https://doi.org/10.1093/zoolinnean/zlaa033

---

## Round 0.2 · Minor Revisions

Dear authors,

Although the manuscript has been widely improved, I still have some minor concerns that I would like that you take into consideration. I even sent the manuscript to a new reviewer to evaluate the current structure of the article and the language, and you can revise the annotated pdf that was submitted to apply the editions suggested.
As a requirement from me is the inclusion of a new figure to show the anatomy and color pattern of the cloacal area for the four found specimens that you consider as belonging to the H. sethmacfarlanei or at least in the three individuals that you collected, although you mentioned that the small juvenile also displayed a similar condition, so, it is possible that you documented the characters in the field (e.g., photographs).

My other suggestion is that you give some attention to the causes of physiological changes in the color and color patterns, mainly derived from changes in environmental patterns. You may have an extensive bibliography about this topic, but I recommend that you include some comments based on the following referenced paper:

Kang, Ch., Kim, Y.E. & Jang, Y.2016. Colour and pattern change against visually heterogeneous
backgrounds in the tree frog Hyla japonica. Scientific Reports 6:22601.DOI: 10.1038/srep22601 www.nature.com/scientificreports

In figure 11, “H.tapichalaca” should be in cursive.

Other comments and edits can be found in the annotated pdf that I am providing.

I hope you will find the new revisions interesting, so I look forward to seeing a new version of your manuscript soon.

Best regards,

Graciela Piñeiro

Reviewer 4 ·

Basic reporting

The manuscript is well written and the literature references are relevant for understanding the description of the new species in the context of the genus. Figures are needed to understand the species description.

Experimental design

Primary research in the areas of systematics, taxonomy, and evolutionary biology

Validity of the findings

The authors compare and differentiate the new species with previous described species in this particular species group.

Annotated reviews are not available for download in order to protect the identity of reviewers who chose to remain anonymous.

---

## Round 0.3 · Minor Revisions

Dear authors,
The manuscript is almost ready, but I have found a few more issues that should be addressed.

Among the most important are the following:

1-You should consider that you are working on extant animals, thus, different than what occurs in Paleontology, it is expected that all the most available information is used to define a new species. At the current stage of your knowledge about these frogs that you are studying is not completely tested in a reasonable sample, you have to use terms that will cover you if other future studies found contradictory results respectful from what you are stated in this article. Therefore, to avoid this potential problem, you should use the term “suggest” or “we can assume” or other similar, that would leave you space for future modifications. Take into account that you only have three or eventually four specimens and you analyzed just two for molecular congruence and just one for internal anatomy (I really would like to know why you did not analyze the internal anatomy of the second specimen, a suggested juvenile that was also euthanized) but it is okay if you will give knowledge to additional information in a forthcoming paper.

2-Caution also applies also for the genetic distance and evolutionary divergences, considering that the group where H. sethmacfarlanei is included has a low bootstrap support and furthermore, you obtained these results from genetic evaluation of only two individuals, a result that may change if you add more individuals in the future. Thus, it is fine to include all your results, but avoid being categorical.

Other minor requests:

3-Please include information in the caption of figure 12 about what is the anatomical position of the scanned skulls and fix H. tapichalaca that should be in cursive.

4-Thanks for including a figure (Fig. 9) for comparison of the cloacal morphology in the four H. sethmacfarlanei individuals, but please provide a scale bar for all images.

I am providing also a new annotated pdf with commentaries as a guide.

Best wishes,
Graciela Piñeiro

---

## Round 0.4 · accepted · Accept

Dear authors,

After reading carefully the last versión of your manuscript I found it more convincing with the complementary explanations made about the methodology followed to calculate the estimation of the divergence time range and genetic distance.

In Paleontology the description of a new species is often based on one individual or worse, on only part of the fossilized remains. So, for us it is expected that a new extant species is defined by many individuals and so, the intraspecific variation is well established. Well, perhaps that was for the beginning my major concern about this manuscript and my commentaries and requests were directed to reach the greatest possible degree of confidence in species identification and also to encourage you (and eventually other colleagues) to continue this study and “fill in the blanks”.

Complementing the information about the applied methodology will give you more chances to revise or highlight some of your results and maybe expand the scope of your study.

For these reasons I decided to accept this current version of your article with the hope that it will encourage the search for new H. sethmacfarlanei individuals to complement in situ for instance the ontogenetic study (including both morphology and change of color) and some paths of its biology (such as for instance the degree of toxicity, call advertising, etc.). Given the apparently reduced population, these studies should be done with caution, preferably in their wild environment.

Congratulations and thanks for having considered most of the suggestions from reviewers and editors in order to improve your manuscript.

Best regards,
Graciela Piñeiro